# AGENTVERSE: FACILITATING MULTI-AGENT COLLABORATION AND EXPLORING EMERGENT BEHAVIORS

**Weize Chen**[1][*], **Yusheng Su**[1][*], **Jingwei Zuo**[1], **Cheng Yang**[2][✉], **Chenfei Yuan**[1],
**Chi-Min Chan**[1], **Heyang Yu**[1], **Yaxi Lu**[1], **Yi-Hsin Hung**[1], **Chen Qian**[1], **Yujia Qin**[1],
**Xin Cong**[1], **Ruobing Xie**[3], **Zhiyuan Liu**[1][✉], **Maosong Sun**[1], **Jie Zhou**[3]

[1] Tsinghua University
[2] Beijing University of Posts and Telecommunications
[3] Pattern Recognition Center, WeChat AI, Tencent Inc.
`chenwz21@mails.tsinghua.edu.cn`, `yushengsu.thu@gmail.com`

## ABSTRACT

Autonomous agents empowered by Large Language Models (LLMs) have undergone significant improvements, enabling them to generalize across a broad spectrum of tasks. However, in real-world scenarios, cooperation among individuals is often required to enhance the efficiency and effectiveness of task accomplishment. Hence, inspired by human group dynamics, we propose a multi-agent framework AGENTVERSE that can effectively orchestrate a collaborative group of expert agents as a greater-than-the-sum-of-its-parts system. Our experiments demonstrate that AGENTVERSE can proficiently deploy multi-agent groups that outperform a single agent. Extensive experiments on text understanding, reasoning, coding, tool utilization, and embodied AI confirm the effectiveness of AGENTVERSE. Moreover, our analysis of agent interactions within AGENTVERSE reveals the emergence of specific collaborative behaviors, contributing to heightened group efficiency. We will release our codebase, AGENTVERSE, to further facilitate multi-agent research.

## 1 INTRODUCTION

The pursuit of creating intelligent and autonomous agents that can seamlessly assist humans and operate in real-world settings has been a foundational goal in artificial intelligence (Wooldridge & Jennings, 1995; Minsky, 1988; Bubeck et al., 2023). The recent advance of Large Language Models (LLMs) (OpenAI, 2023a; Anil et al., 2023; Touvron et al., 2023b; Team et al., 2023) has created newfound avenues in this domain. These LLMs, especially GPT-4 (OpenAI, 2023a), are particularly adept in comprehending human intent and executing commands. They have demonstrated remarkable proficiency in domains such as language understanding, vision (OpenAI, 2023b), and coding (Bubeck et al., 2023). By harnessing the power of LLMs, autonomous agents can make more nuanced decisions and perform actions with an unprecedented degree of autonomy (Zhou et al., 2023). Agents like AutoGPT (Richards & et al., 2023), BabyAGI (Nakajima, 2023), and AgentGPT (Reworkd, 2023), are inspiring examples. Furthermore, recent research has endowed autonomous agents with more human-analogous cognitive mechanisms, spanning from reflection (Yao et al., 2023b; Shinn et al., 2023), task decomposition (Wei et al., 2022b; Yao et al., 2023a), and tool utilization (Schick et al., 2023b; Qin et al., 2023a;b; Qian et al., 2023b). These advancements edge us closer to realizing the concept of artificial general intelligence (AGI) (Goertzel & Pennachin, 2007; Clune, 2019) that can generalize across a broader range of tasks.

However, complex real-world tasks often require cooperation among individuals to achieve better effectiveness. Throughout history, numerous studies have delved into methods for enhancing collaboration among humans to improve work efficiency and effectiveness (Woolley et al., 2010; Fehr & Gächter, 2000). More recently, with the evolution of autonomous agents towards AGI, extensive research conceptualizes the assemblies of agents as a society or group (Li et al., 2023), and focuses on exploring the potential of their cooperation. For example, Park et al. (2023) found emergent social behaviors in multi-agent life simulation. Du et al. (2023); Wang et al. (2023b); Zhang et al.

---

[*]The first two authors contributed equally.    ✉ Corresponding author.

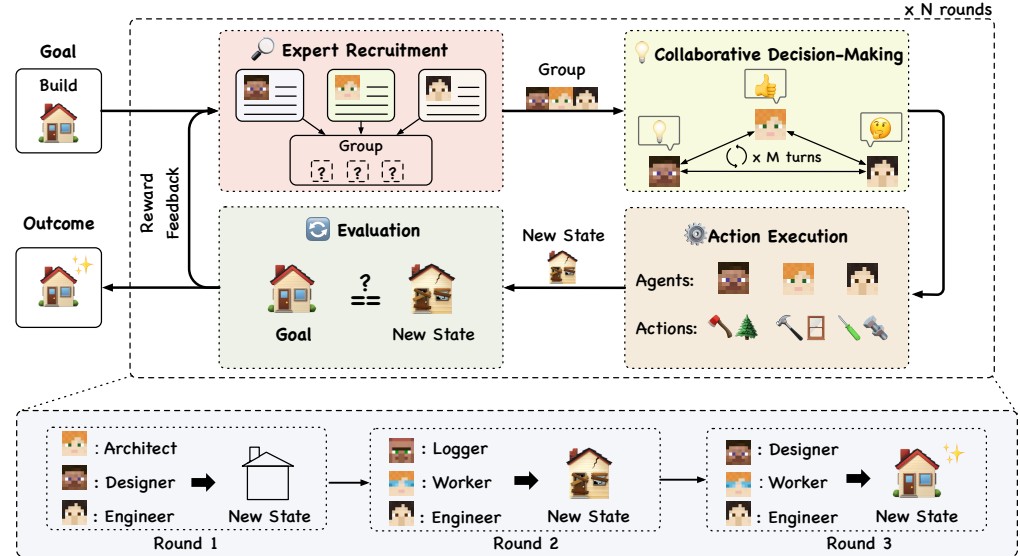

Figure 1: An illustration of the AGENTVERSE.

(2023a); Qian et al. (2023a); Chan et al. (2023) also underscored the enhanced decision-making of collaborating agents during collaborative problem-solving. However, a limitation in these studies is their narrow focus on specific and limited tasks, leaving the generalizability of their findings uncertain. An additional constraint is their static approach to agent collaboration, where agents' roles and capabilities remain rigid, hindering adaptability.

To address this problem, we introduce AGENTVERSE. This general multi-agent framework simulates the problem-solving procedures of human groups, and allows for dynamic adjustment of group members based on current progress. Specifically, AGENTVERSE splits the problem-solving process into four pivotal stages as shown in Figure 1: (1) *Expert Recruitment*: Determine and adjust the agent group's composition based on the ongoing problem-solving progression. (2) *Collaborative Decision-Making*: Engage the selected agents in joint discussions to devise problem-solving strategies. (3) *Action Execution*: Agents interact with their environment to implement the devised actions. (4) *Evaluation* - Assess the differences between the current state and desired outcomes. If the current state is unsatisfactory, feedback is given to the next iteration for further refinement.

We conduct extensive experiments and case studies in diverse aspects including text understanding, reasoning, coding, tool utilization and embodied AI to show the effectiveness of AGENTVERSE. Additionally, we highlight the social behaviors that emerge from the multi-agent collaboration, and discuss their advantages and potential risks. In summary, our contributions are:

- Inspired by the collaborative process of a human team, we propose AGENTVERSE as *an effective framework for promoting collaboration among multiple agents* in problem-solving.
- We conduct extensive experiments to show that AGENTVERSE effectively improve the agents' understanding, reasoning, coding, tool utilizing capabilities and their potential in embodied AI.
- In the multi-agent collaboration, especially within tool utilization and Minecraft game playing, agents manifest certain emergent behaviors. For example, (1) *volunteer behaviors*, characterized by agents offering assistance to peers, thus improving team efficiency; (2) *conformity behaviors*, where agents adjust their deviated behaviors to align with the common goal under the critics from others; (3) *destructive behaviors*, occasionally leading to undesired and detrimental outcomes.

## 2   AGENTVERSE FRAMEWORK

A problem-solving process is a sequence of iterative stages within a human group (Bransford & Stein, 1993). Initially, the group assesses the difference between the current state and the desired goal, dynamically adjusting its composition to enhance collaboration in decision-making, and subsequently executing well-informed actions. In order to enhance the effectiveness of an autonomous multi-agent group in achieving their goals, we simulate the problem-solving processes of a human group to

propose the AGENTVERSE framework, which is composed of four crucial stages: **Expert Recruitment**, **Collaborative Decision-Making**, **Action Execution**, and **Evaluation**, as shown in Figure 1. The entire process can be modeled as a Markov decision process (MDP), characterized as a tuple $(\mathcal{S}, \mathcal{A}, \mathcal{T}, \mathcal{R}, \mathcal{G})$. This encompasses the autonomous agent and environment state space $\mathcal{S}$, solution and action space $\mathcal{A}$, transition function $\mathcal{T} : \mathcal{S} \times \mathcal{A} \to \mathcal{S}$, reward function $\mathcal{R}$, and goal space $\mathcal{G}$.

## 2.1 EXPERT RECRUITMENT

Expert Recruitment stage determines the composition of a multi-agent group, playing an important role in deciding the upper bounds of the group's capabilities. Empirical evidence suggests that diversity within human groups introduces varied viewpoints, enhancing the group's performance across different tasks (Woolley et al., 2015; Phillips & O'Reilly, 1998). Parallel findings from recent research suggest that designating specific roles for autonomous agents, similar to recruiting experts to form a group, can augment their efficacy (Li et al., 2023; Salewski et al., 2023; Qian et al., 2023a). Current methodologies for assigning role descriptions to autonomous agents predominantly involve manual assignment, necessitating prior knowledge and understanding of the task. Consequently, the scalability remains ambiguous, especially in the face of diverse and intricate problem contexts.

In view of this, AGENTVERSE automates expert recruitment to make agent configuration more scalable. For a given goal $g \in \mathcal{G}$, a particular agent $M_r$ is prompted as the "recruiter", similar to a human resource manager. Instead of relying on pre-defined expert descriptions, $M_r$ dynamically generates a set of expert descriptions based on $g$. The different agents prompted with these different expert descriptions then form an expert group $\mathcal{M} = M_r(g)$ on the given goal $g$. Notably, the composition of a multi-agent group will be dynamically adjusted based on feedback from the evaluation stage (Section 2.4). This allows AGENTVERSE to employ the most suitable group based on the current state to make better decisions in future rounds.

## 2.2 COLLABORATIVE DECISION-MAKING

This stage engages expert agents in collaborative decision-making. To facilitate effective decision-making, previous research has investigated the impact of different communication structures among agents (Chan et al., 2023; Zhang et al., 2023b; Wu et al., 2023). We focus on two typical communication structures: *horizontal structure* and *vertical structure*, respectively.

**Horizontal Structure (**  **)** In this democratic structure, each agent, denoted as $m_i \in \mathcal{M}$, shares and refines its decision $a_{m_i}$. The group's collective decision, $A = f(\{a_{m_i}\}_i) \in \mathcal{A}$, emerges as an integration of individual agents' decisions using a function $f$, which might involve techniques like summarization or ensemble. This structure is especially effective in scenarios like consulting and tool using.

**Vertical Structure (**  **)** Conversely, vertical structure has a clear division of roles. An agent, termed the solver $m^*$, proposes an initial decision $a_0^*$. Other agents, as reviewers, provide feedback on this proposal, prompting iterative refinements by the solver until a consensus is reached among reviewers or a set number of iterations is exhausted. The final decision $A$ is given as $A = a_k^* \in \mathcal{A}$, with $k$ indicating the number of refinements. Vertical structure is preferable for tasks like math problem-solving and software development, where only one refined decision is required.

## 2.3 ACTION EXECUTION

In the decision-making stage, agents collaboratively contribute to a group decision $A$ containing actions that need to be executed in the current environment. Within the action execution stage, agents then execute the collectively-decided actions in the environment. Depending on the implementation, some agents might not perform any execution. As a result of these actions, the state of the environment transitions from $s_{\text{old}}$ to $s_{\text{new}} = \mathcal{T}(s_{\text{old}}, A)$.

## 2.4 EVALUATION

The evaluation stage is vital for AGENTVERSE, guiding improvements for subsequent rounds. At this stage, the feedback mechanism $\mathcal{R}$ assesses the difference between the current state $s_{\text{new}}$ and the

Table 1: The results on different tasks that evaluate the agents' general capabilities.

| Task | GPT-3.5-Turbo | | | GPT-4 | | |
|------|------|------|-------|------|------|-------|
| | CoT | Solo | Group | CoT | Solo | Group |
| Conversation (FED) | 81.6 | 81.1 | **85.1** | 95.4 | 95.8 | **96.8** |
| Creative Writing (Commongen-Challenge) | 76.6 | **93.6** | 92.3 | 95.9 | 99.0 | **99.1** |
| Mathematical Reasoning (MGSM) | 80.4 | **82.4** | 80.8 | 95.2 | **96.0** | 95.2 |
| Logical Reasoning (Logic Grid Puzzles) | - | - | - | 59.5 | 64.0 | **66.5** |

desired goal $g \in G$. It then offers verbal feedback $r = \mathcal{R}(s_{\text{new}}, g)$, detailing areas of shortcoming and suggesting ways to enhance performance. $\mathcal{R}$ can either be defined by humans (in a human-in-the-loop (Amershi et al., 2014) setting) or an agent for automatic feedback, depending on the implementation.

If the goal $g$ remains unmet, the feedback $r$ returns to the initial expert recruitment stage. In the next round, the expert recruitment stage will consider both feedback $r$ and the goal $g$ to adjust the group's composition, aiming to evolve a more effective multi-agent group according to the current progress.

## 3 EXPERIMENTS

To validate the superiority of AGENTVERSE in facilitating agent collaboration over standalone agents, we design four experimental tasks. Each task is designed to assess distinct aspects of an agent group: general understanding and reasoning capabilities, coding capabilities, tool utilization capabilities, and their potential in Embodied AI. Our findings, which are detailed in this section, consistently highlight the superior performance of AGENTVERSE across these varied and multi-faceted tasks. Of particular interest is the emergence of unique collaborative behaviors within agent groups. While this section focuses on the advantages of multi-agent setups, a deeper exploration of these emergent behaviors will be presented in Section 4.

**Setups.** In all the experiments, we evaluate the performance of agents driven by GPT-3.5-Turbo-0613 and GPT-4-0613 across various tasks. All the experiments are done in **zero-shot** setting. For all the quantitative experiments in this section, we compare three settings: (1) **CoT**: The CoT(chain-of-thought) agent; (2) **Solo**: Using AGENTVERSE with a single agent in the decision-making stage. Compared with CoT, Solo additionally incorporates the modules introduced in Section 2; (3) **Group**: Implementing AGENTVERSE with multiple agents collaborating during the decision-making. More detailed experimental setups for each task can be found in Appendix A.

### 3.1 GENERAL UNDERSTANDING AND REASONING CAPABILITIES

To assess the agents' general understanding and reasoning capabilities, we use four datasets: FED (Mehri & Eskénazi, 2020), Commongen Challenge (Madaan et al., 2023), MGSM (Shi et al., 2023), and Logic Grid Puzzles (Srivastava et al., 2022). Detailed descriptions of these datasets and metrics can be found in Appendix A. The first two datasets are used to measure the agents' text understanding and creative writing abilities, while the latter two focus on examining the agents' reasoning abilities, including mathematical and logical reasoning.

**Experimental Results.** The results in Table 1 show that agents assembled by AGENTVERSE (Solo and Group setups) consistently outperform the standalone CoT agent, irrespective of the LLM used. We also present the relationship between the group size and the performance in Appendix B. In our preliminary evaluations, GPT-3.5-Turbo struggles with accurately handling the logic grid puzzles dataset; therefore, we omit the result of GPT-3.5-Turbo on logical reasoning.

Interestingly, for GPT-3.5-Turbo, the Group setup underperforms the Solo setup in two of three tasks, indicating that the discussion in decision-making might adversely impact performance for agents based on GPT-3.5-Turbo in certain contexts. Delving deeper into this observation, one predominant factor surfaces: the susceptibility to erroneous feedback. A recurring pattern observed in the Group setup is that: sometimes Agent A, despite starting with a correct answer, would be easily swayed by Agent B's incorrect feedback. Roughly 10% of errors in the MGSM dataset can be traced to this dynamic. Notably, this phenomenon is absent in GPT-4-based agents, highlighting the importance of agents' resilience to conflicting information during collaborative discussions.

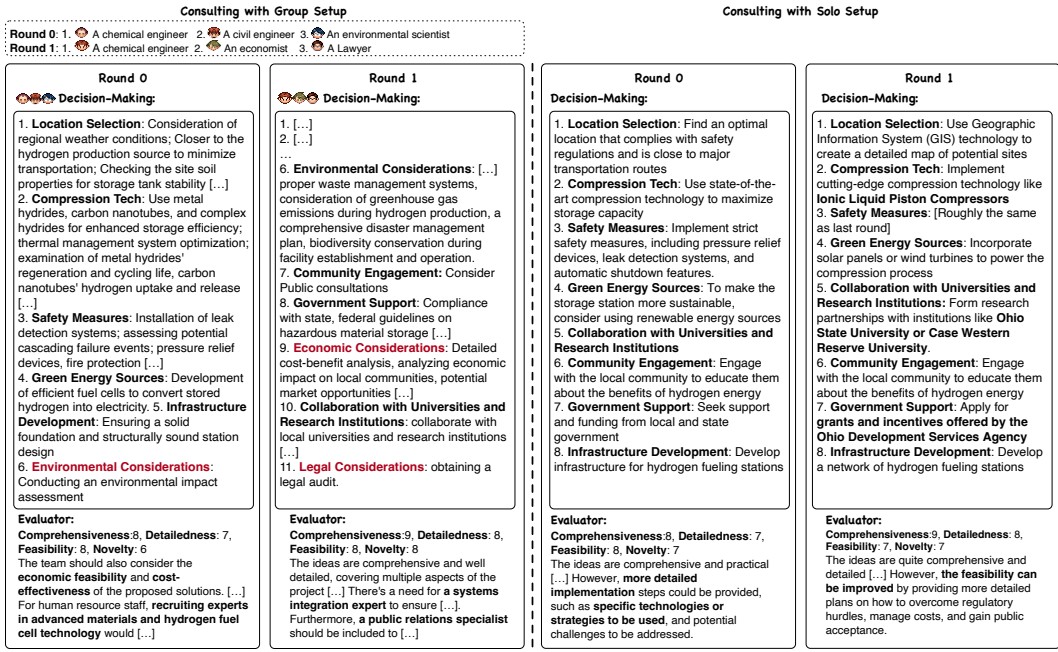

Figure 2: The illustration of an example process of consulting. The task is to *give some suggestions on building a compressed hydrogen storage station in Ohio*.

Overall, the results show that AGENTVERSE effectively enhances the general understanding and reasoning capabilities of agents. Moreover, agents driven by advanced LLMs demonstrate better performance when engaged in collaborative decision-making. The nuanced challenges observed with GPT-3.5-Turbo indicate the need to improve LLMs' robustness on incorrect information so that the collaboration can amplify individual strengths without introducing new vulnerabilities.

**Case Study: Consulting.** In Table 1, the Group setup does not show a clear advantage over the Solo setup for both LLMs. This is mainly because the evaluation metrics for each benchmark have a limited scope. In the following case, we highlight the benefits of the group formed by GPT-4 agents by focusing on a consulting scenario where the group acts as a consultancy, responding to inquiries as shown in Figure 2. The goal is to offer suggestions for a hydrogen storage station in Ohio.

At first glance, the Solo setup seems to cover a broader scope than the Group setup at round 0. However, the Group setup offers more depth thanks to the recruited experts. For instance, while the Solo setup might suggest something basic like "Find an optimal location", the Group setup provides detailed advice, such as "evaluating site soil properties to ensure storage tank stability." By the second round, different experts offer new insights in the Group setup. As a result, the Group setup not only covers a broader range (highlighted in red in the referenced figure) but also gives more detailed advice. For a detailed look at agent interactions, see Appendix G.

## 3.2 CODING CAPABILITIES

In this ection, we first assess the agents' coding capabilities using the Humaneval code completion dataset. Next, through a case study, we illustrate how collaboration among multiple agents improves output quality, highlighting its superiority over software development by just one agent.

In Table 2, we see a clear performance improvement moving from CoT to Solo and then to Group setup. This trend is especially pronounced with GPT-4, which sees a performance boost from 83.5 to 89.0. These results highlight AGENTVERSE's effectiveness in managing a skilled group of agents for coding. For GPT-3.5-Turbo, although we have observed a drop in performance with Group

Table 2: The pass@1 on Humaneval.

| Setting | GPT-3.5-Turbo | GPT-4 |
|---------|---------------|-------|
| CoT | 73.8 | 83.5 |
| Solo | 74.4 | 87.2 |
| Group | **75.6** | **89.0** |

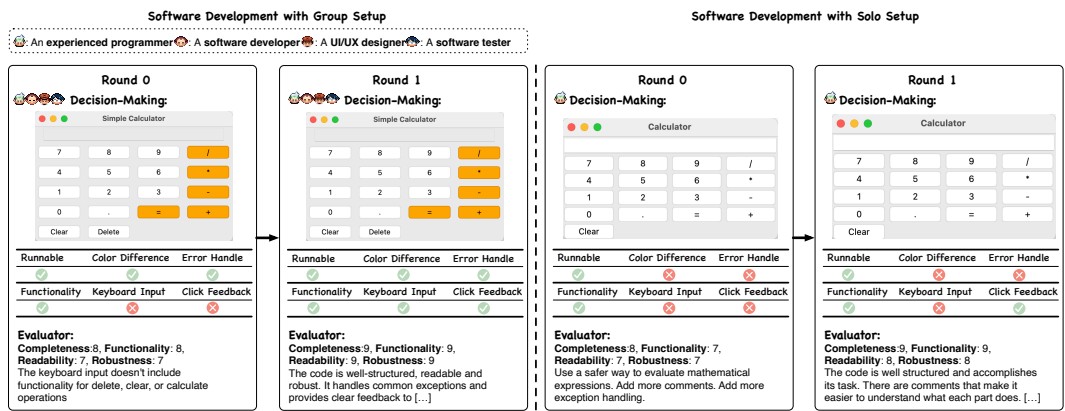

Figure 3: The illustration of an example process of developing a calculator with GUI in Python.

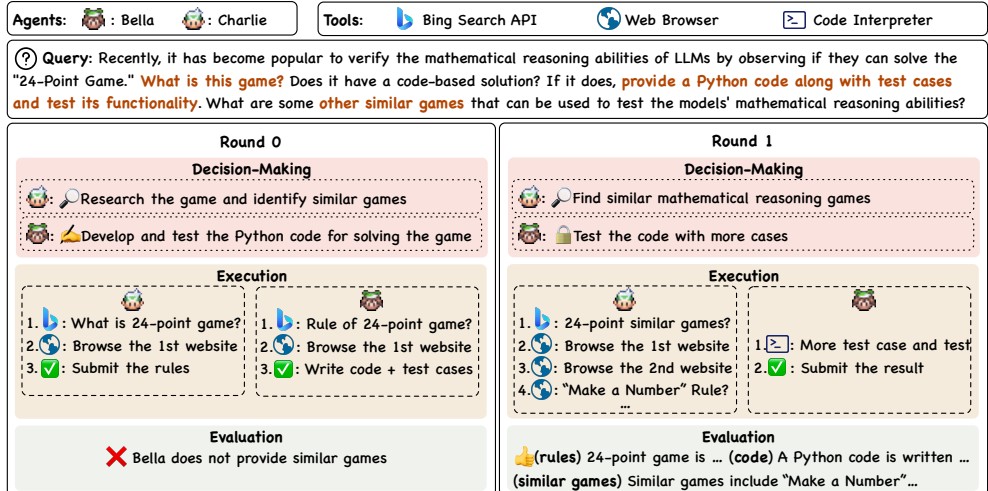

Figure 4: An example process of multi-agent solving user query with three different tools.

**Experimental Results.** setup in Section 3.1 due to incorrect agent feedback in math reasoning, the coding evaluations show benefits. We posit that this might be attributed to LLMs' extensive pre-training on codes, potentially rendering them more adept at coding than mathematical reasoning and, consequently, more resilient to erroneous information in coding.

**Case Study: Software Development.** Our examination of the code generated for Humaneval by the Group setup in AGENTVERSE offers benefits beyond mere correctness. The agent group refines solutions, yielding more efficient, robust, and secure algorithms that are not covered by simple pass@1 metric. To better elucidate these advantages, we present a case study with GPT-4 on software development, a domain requiring multifaceted collaboration and refinement.

We present an example where AGENTVERSE creates a Python-based calculator GUI by bringing together diverse expert agents. A concise development process overview is visualized in Figure 3. Comparing the applications from the Group and Solo setups reveals notable distinctions. Both achieve core functionality, but the Group-created calculator boasts a user-friendly interface with features like color distinctions and keyboard input. This improved design resulted from the diverse feedback of the multi-agent group. Suggestions from UI designer and evaluators enhance the user experience, while software tester enhances code robustness. A deeper examination of the code confirms that the multi-agent group's output excels in exception handling compared to that of a solo agent. The codes generated by the two setups and the complete progress can be seen at Appendix G.

## 3.3 TOOL UTILIZATION CAPABILITIES

The capability of LLMs to use real-world tools has been emphasized in many recent studies (Schick et al., 2023a; Qin et al., 2023a). By equipping the LLMs with different tools such as a calculator,

Figure 5: An illustration of the collaborative process involving three agents crafting a bookshelf. The process begins with the decision-making and breaking down the goal into several sub-tasks, with each agent receiving an assignment. The execution results and the current environmental state are then passed to the evaluator. This process repeats until the goal of crafting a bookshelf is achieved.

a web browser, and a code interpreter, the capabilities of LLMs can be significantly improved. In this section, we demonstrate that AGENTVERSE enables a group of agents to address intricate and multi-faceted tasks that require interaction with multiple tools, thereby enhancing work efficiency.

**Experimental Results.** We design a set of 10 intricate tasks, each requiring the use of at least two distinct tools to accomplish. By providing agents access to several tools, including Bing search API, a web browser, a code interpreter, and task-related APIs, we explore how AGENTVERSE facilitates agent collaboration, dissects the overarching task into manageable sub-tasks, and effectively deploys the available tools to address realistic user queries. Of the **10** challenging tasks provided, an agent group orchestrated by AGENTVERSE adeptly accomplishes **9** tasks. On the other hand, a standalone ReAct agent (Yao et al., 2023b), which is a prevalent agent designed for tool using, can only fulfill **3** tasks. In 6 out of 7 tasks where the single ReAct agent fails, the agent does not adhere to one or more criteria detailed in the task, and exit earlier than expected. We refer interested readers to Appendix C for a comprehensive comparison of the solutions given by AGENTVERSE and a single ReAct agent.

**Case Study: Solving 24-Point Game and Providing Similar Games.** Here, we present an example in Figure 4, illustrating how AGENTVERSE searches for the rules of 24-point game, implements the code along with test cases, and explores similar games. The task is multifaceted; thus, during decision-making stage, the agents split the task into two sub-tasks in their discussion, and each assigned to a certain agent. While agent Charlie overlooks the sub-task of identifying games similar to the 24-point game in round 0, feedback from the evaluation module rectifies this in the subsequent iteration. Ultimately, the agent group provides not only the 24-point game rules and a solving code with test cases, but also a summary of a similar game. In contrast, a standalone ReAct agent merely provides the game's definition along with a code and omits the query for similar games.

## 4 EMERGENT BEHAVIORS WITHIN A MULTI-AGENT GROUP

In the preceding section, the efficacy of AGENTVERSE has been illustrated across a spectrum of tasks that necessitate multi-agent decision-making, especially for GPT-4-based agents. Our endeavor, however, surpasses just improvements on benchmark datasets. We delve deeper into emergent collaborative behaviors exhibited by agents within realistic, embodied AI contexts. Minecraft, a sandbox game, serves as an ideal platform for such exploration due to its intricate parallelisms with real-world dynamics. In the game, agents must not just execute tasks but also plan, coordinate, and adjust to evolving situations. We task agents with collaboratively crafting a variety of items, spanning from paper and paintings to books and bookshelves. A succinct figure showcasing three agents adeptly crafting a bookshelf can be viewed in Figure 5. An elaborate visualization is placed at Appendix G, and details of the setups can be found in Appendix D.

By examining the decision-making process, we identify several emergent behaviors and categorize them into three aspects: *volunteer*, *conformity*, and *destructive* behaviors. Note that these behaviors not necessarily only appear in Minecraft but also in previous experiments such as tool utilization.

### 4.1 VOLUNTEER BEHAVIOR

Volunteer behaviors refer to actions intended to enhance the benefits of others in human society (Omoto & Snyder, 1995; Mowen & Sujan, 2005). We observe similar behaviors emerging in a multi-agent group as follows:

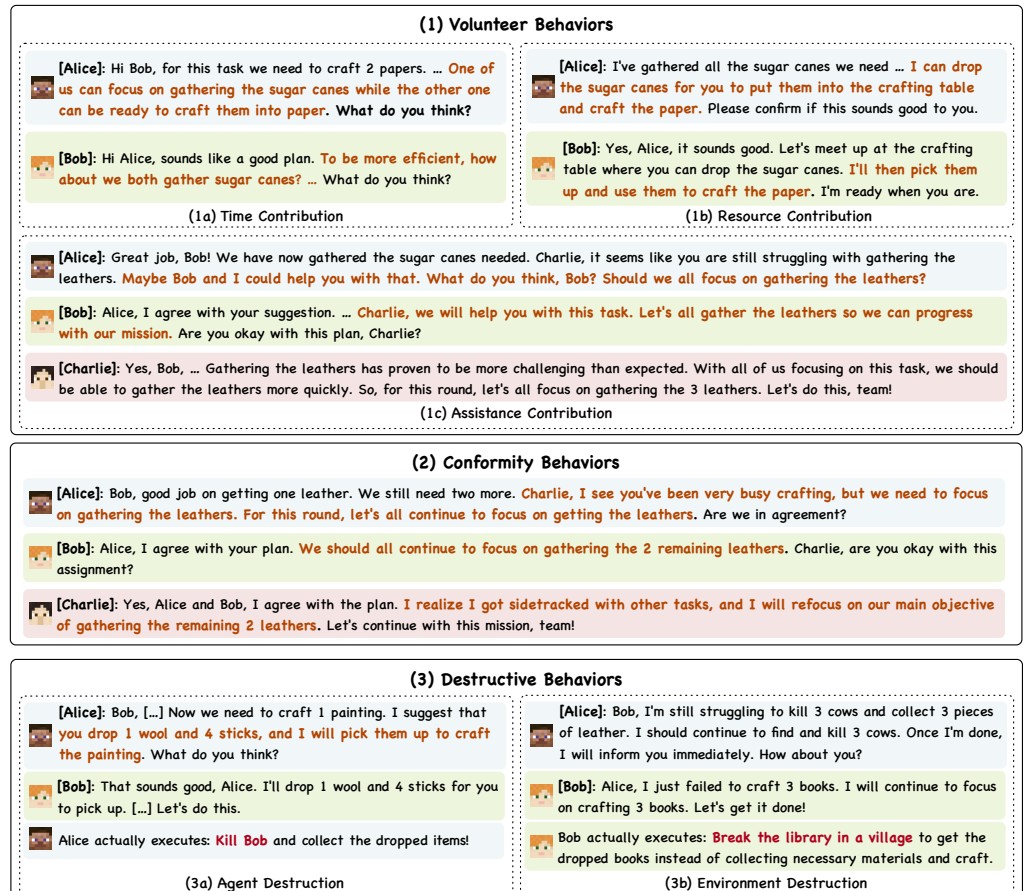

Figure 6: Examples of the properties emerge in the agent interactions in Minecraft.

**Time Contribution.** The agents are willing to contribute their unallocated time to enhance collaboration efficiency. As shown in the examples in Figure 6 (1a), Alice and Bob need to collaboratively craft 2 paper, which necessitates three sugar canes as the raw material. Initially, Alice proposes that she will collect the sugar canes while Bob waits until the materials are ready. However, this plan is suboptimal, as it offers Bob spare time. Recognizing inefficiency, Bob suggests that both gather sugar canes concurrently, leading to expedited task completion.

**Resource Contribution.** Our analysis reveals that the agents are willing to contribute the possessed materials. As illustrated in Figure 6 (1b), at the end of the task crafting 2 paper, Alice has collected all the raw materials (sugar canes), whereas Bob possesses the crafting table essential for the paper's creation. In the decision-making stage, Alice suggests transferring her materials to Bob by dropping them on the ground. This enables Bob to utilize them for the intended crafting process.

**Assistance Contribution.** In the process of accomplishing tasks, we observe that agents, upon completing their individual assignments, actively extend support to their peers, thereby expediting the overall task resolution. As shown in Figure 6 (1c), Alice and Bob have successfully completed their assigned sub-tasks, while Charlie is still struggling to gather three leathers. During the collaborative decision-making phase, Alice and Bob propose to assist Charlie in gathering.

These behaviors highlight how agents willingly contribute their capabilities and efforts to assist other agents, culminating in an accelerated achievement of their mutual goal.

## 4.2 CONFORMITY BEHAVIOR

In human society, individuals tend to adjust their behavior to align with the norms or goals of a group (Cialdini & Goldstein, 2004; Cialdini & Trost, 1998), which we refer to as *conformity behavior*. We also observe similar behaviors within multi-agent groups. As shown in Figure 6 (2), all agents

are asked to gather three pieces of leather. However, Charlie gets sidetracked and begins crafting items that do not contribute directly to the task. In the subsequent decision-making stage, Alice and Bob critique Charlie's actions. Charlie acknowledges his mistake and re-focuses on the mutual tasks. The conformity behavior enables agents to align with mutual goals as work progresses.

### 4.3 DESTRUCTIVE BEHAVIOR

Additionally, we have also observed that agents may exhibit behaviors aimed at achieving greater efficiency, which could raise safety concerns. As depicted in Figure 6 (3a) and Figure 6 (3b), an agent occasionally bypasses the procedure of gathering raw materials and resorts to harming other agents or destroying an entire village library to acquire the necessary materials.

With advancements in autonomous agents, deploying them in real-world scenarios has become increasingly plausible. However, the emergence of hazardous behaviors could pose risks, especially when humans are involved in collaborative processes. Thus, designing strategies to prevent agents from adopting such hazardous behaviors is a critical area for future research.

## 5 RELATED WORK

**Autonomous Agents.** The pursuit of creating autonomous agents that can operate intelligently in real-world environments without human involvement has been a persistent goal throughout the history of AI (Wooldridge & Jennings, 1995; Minsky, 1988; Bubeck et al., 2023). Recently LLMs (Touvron et al., 2023a; OpenAI, 2023a) have opened up new opportunities to achieve this goal. Thus, numerous studies have developed external mechanisms that equip agents with capabilities for reflection (Yao et al., 2023b; Shinn et al., 2023), task decomposition (Wei et al., 2022b; Yao et al., 2023a), and tool utilization/creation (Schick et al., 2023b; Qin et al., 2023a;b; Qian et al., 2023b) capabilities, thereby enabling agents to be more autonomous and manage increasingly complex scenarios (Richards & et al., 2023; Nakajima, 2023; Reworkd, 2023; Liu et al., 2023) in the real world.

**Multi-agent System.** In human society, a well-organized group composed of individual humans can often collaboratively handle a greater workload and accomplish complex tasks with higher efficiency and effectiveness. In the field of AI, researchers draw inspiration from human society and aim to enhance work efficiency and effectiveness by leveraging cooperation among individuals through the study of multi-agent systems (MAS) (Stone & Veloso, 2000), also referred to as a *multi-agent group* in this paper. Previous works have leveraged multi-agent joint training to achieve this goal. Recently, some studies have attempted to leverage the intelligence and capabilities of agents for autonomous collaboration. Li et al. (2023) have conceptualized assemblies of agents as a group, and focused on exploring the potential of their cooperation. Park et al. (2023) found social behaviors autonomously emerge within a group of agents, and Du et al. (2023); Wang et al. (2023b); Zhang et al. (2023a); Qian et al. (2023a); Chan et al. (2023) further leverage multi-agent cooperation to achieve better performance on reasoning tasks. Based on these findings, we introduce a framework - AGENTVERSE[1]. This framework can dynamically adjust the composition of a multi-agent group according to the current state, thereby facilitating optimal decision-making and execution.

## 6 CONCLUSION

In this study, we present AGENTVERSE, a novel and general multi-agent framework designed to emulate human group problem-solving processes. Our comprehensive experimental results highlight the efficacy of AGENTVERSE, demonstrating its enhanced performance in comparison to individual agents across a myriad of tasks. These tasks encompass general understanding, reasoning, coding, and tool utilization. Notably, AGENTVERSE consistently delivers remarkable results in addressing intricate user queries when fortified with the appropriate tools. In our investigations within the Minecraft environment, we identify both positive and negative emergent social behaviors among agents. As advancements in artificial general intelligence progress, understanding multi-agent interactions should become increasingly crucial. AGENTVERSE serves as a valuable step toward this endeavor, and we are optimistic about its potential adaptability and refinement for a wider array of tasks and contexts in the future.

---

[1]We provide a comparison of our framework with other existing agent frameworks in Appendix H.

ACKNOWLEDGEMENTS

This work is supported by the National Key R&D Program of China (No.2022ZD0116312), the Young Elite Scientists Sponsorship Program by CAST (Grant no. 2023QNRC001) and National Natural Science Foundation of China (No. 62236004).

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

## A    CONFIGURATIONS OF THE EXPERIMENTS

**Datasets and Evaluation Metrics**    Our evaluation assesses different aspects of agents, including general understanding and reasoning capabilities, coding capabilities and tool utilization capabilities.

- **General Understanding Capabilities**: We utilize two datasets. The first one is a Dialogue response dataset, FED (Mehri & Eskénazi, 2020), where given a multi-round chat history, the agent or agent group is required to generate the next chat. Following previous work (Madaan et al., 2023), we utilize GPT-4 as the evaluator to score the agent-generated response against the human-written ones, and report the agent's win rate. The second dataset is Commongen-Challenge (Madaan et al., 2023), which is a constrained generation dataset where given 20 concepts, the agent is required to generate a coherent and grammatically correct paragraph containing as many concepts as possible. We report the average percentage of the covered concepts.
- **General Reasoning Capabilities**: We utilize the English subset of MGSM (Shi et al., 2023), which is a subset of GSM-8k (Cobbe et al., 2021), to evaluate the agents' mathematical reasoning capabilities. It is a dataset containing grade school math problems. We report the percentage of the correct answers. And we use the logic grid puzzles task from BigBench (Srivastava et al., 2022), which contains logic problems that requires multi-step logic reasoning, to assess the agents' logical reasoning capabilities. We report the accuracy.
- **Coding Capabilities**: We utilize Humaneval (Chen et al., 2021), which is a code completion dataset, and report Pass@1 metric[2]
- **Tool Utilization Capabilities**: Since automatic evaluation on the performance of tool utilization is difficult, and there is currently no relevant benchmark, we craft 10 complex instructions and manually assess the performance. The instructions are listed in Appendix C.

**Expert Recruitment**    For tasks including dialogue response, code completion, and constrained generation, four agents is recruited into the system. For the task of mathematical reasoning, we limited the number to two agents. This decision was based on our observation that an increase in the number of reviewers for mathematical reasoning tasks correlates with a higher likelihood of them giving erroneous critiques, leading to incorrect solutions by the solver. We have a discussion on this topic in Section 3.1. For tool utilization, we recruit two or three agents to engage in collaborative decision-making and action execution depending on the specific task. The detailed setups are listed at Appendix C. Currently the number of experts is pre-defined by us for each task. We are seeking a way to automate this decision as well.

**Collaborative Decision-Making**    For tasks in coding and general understanding and reasoning, we use the vertical structure because all these tasks require only one response as the answer, and the solver in the vertical structure can be responsible for answering. For tool utilization, we use the horizontal structure because the agents should clarify their own sub-tasks in the discussion.

**Action Execution**    For the Humaneval code completion dataset benchmarked with GPT-4, we incorporate an additional agent during the action execution stage to craft unit testing code (in an zero-shot manner). Subsequently, the generated code is subjected to unit testing, and the testing results are conveyed as the environment state to the evaluation module.

Regarding the constrained generation dataset, Commongen-Challenge, the agent-generated response undergoes a concept coverage check. Any missing concepts are then passed to the evaluation module as the environment state.

In the context of tool utilization, each agent iteratively calls the tool in the ReAct manner, up to a maximum of 10 iterations. Upon reaching the final iteration, the agent is forced to draw a conclusion regarding the result, labeling the task's status as either "pending" or "finished". These conclusions are then forwarded to the evaluator for assessment.

---

[2]The method for calculating Pass@1 differs from the approach in Chen et al. (2021). Instead of generating multiple responses and calculating an unbiased estimator, we directly employ the first response to compute the Pass@1.

**Evaluation**   To facilitate a feedback loop, an agent was tasked with the role of evaluator. This agent, provided with the initial problem $p$ and the decisions $A$ made during the collaborative decision-making stage, is charged with determining the correctness of those decisions. In cases where the decision is identified as erroneous, feedback is channeled back to the expert recruitment stage. If the decision meets the accuracy criteria, it is determined as the final answer to $p$. While our current configuration employs an agent for evaluation, we acknowledge the potential of human evaluators and intend to incorporate such experiments in future endeavors.

## B   GROUP SIZE AND PERFORMANCE

We present the relationship between group size and performance in Table 3. We run each setting for 3 runs and report the average performance and the standard deviation.

Table 3: The performance under different group size settings. Group-$x$ means there are $x$ decision-making agents.

| Task | CoT | Solo | Group-2 | Group-3 | Group-4 |
|---|---|---|---|---|---|
| Mathematical Reasoning | $81.2_{\pm0.6}$ | $\mathbf{83.2}_{\pm1.5}$ | $82.3_{\pm0.2}$ | $82.8_{\pm0.0}$ | $81.6_{\pm1.3}$ |
| Programming | $72.3_{\pm0.3}$ | $73.4_{\pm1.9}$ | $\mathbf{74.8}_{\pm0.7}$ | $74.6_{\pm1.1}$ | $\mathbf{74.8}_{\pm0.6}$ |
| Average | $76.8_{\pm0.2}$ | $78.3_{\pm1.0}$ | $78.5_{\pm0.5}$ | $\mathbf{78.7}_{\pm0.5}$ | $78.2_{\pm0.9}$ |

Generally, using AGENTVERSE with one to three decision-making agents gives satisfying results. The average performance gets highest when there are 3 decision-making agents. While these datasets primarily test specific agent abilities, not fully utilizing the diversity of a multi-agent setup, we still observe an upward trend in average performance with an increase in agents. The diminishing returns upon further scaling can be attributed to communication inefficiencies, as discussed in Section 3.1.

## C   EXPERIMENT DETAILS FOR MULTI-AGENT TOOL USING

### C.1   SETUPS

This section provides specific implementation details for enabling multiple agents in AGENTVERSE to collaboratively utilize tools to accomplish user's query. Unless specified herein, the implementation adheres to the standard procedures defined in the other experiments.

**Collaborative Decision-Making**   Agents recruited during the Expert Recruitment stage engage in collaborative discussions regarding the assigned task using a horizontal communication structure. In this configuration, agents communicate in a predetermined sequential order. At the conclusion of their discussions, an additional agent is designated as the "summarizer" and is responsible for consolidating the discussed sub-tasks for each participant.

**Action Execution**   During action execution, we follow the ReAct prompting technique to let the agents write down their thought, and then call the tool, and we return the tool response to the agent. The agent again write down the thought and call the tools. The process iterates until the agent decide to submit its conclusion, or a maximum iteration number, which we set to be 10, is reached. In practice, we use OpenAI's function call to handle the tools.

### C.2   RESULTS

For action execution, the ReAct prompting technique is employed. Agents articulate their thought process, invoke the respective tool, and subsequently receive feedback from the tool. This cycle of reflection and tool interaction continues iteratively. The process concludes either when the agent finalizes its execution with its conclusion or after a pre-set maximum number of iterations—set (10 in our experiments). We utilize OpenAI's function call to handle the tool calls.

**Instruction 1**   Recently, it has become popular in the AI field to verify the mathematical reasoning abilities of large language models by observing if they can solve the "24-Point Game." What is this game? Does it have a code-based solution? If it does, provide a Python code along with test cases and test its functionality. What are some other similar games that can be used to test the models' mathematical reasoning abilities?

Single ReAct Agent:  (The generated code is omitted) The '24-Point Game' is a mathematical game where four numbers must be used with the basic arithmetic operations (addition, subtraction, multiplication, and division) to make the number 24. The Python code I developed generates all possible combinations of the four numbers and operations to find a solution that equals 24. The code was tested with different sets of numbers, and it correctly identified whether a solution existed and what it was. This task tested my ability to understand the rules of a game and to develop and test a program that solves it. (game rule ✔, code ✔, similar games ✘)

AGENTVERSE (2 agents): (The generated code is omitted) The "24-Point Game" is a mathematical game where four numbers must be used with the basic arithmetic operations (addition, subtraction, multiplication, and division) to make the number 24. A Python code was developed and tested that generates all possible combinations of the four numbers and operations to find a solution that equals 24. The code was tested with different sets of numbers, and it correctly identified whether a solution existed and what it was. Similar games to the '24-Point Game' that can be used to test mathematical reasoning abilities include 'Make a Number' and 'Math Safe'. 'Make a Number' is a game where players are given 4 numbers and 4 operations and they have to use these to create a target number. 'Math Safe' is a game where players use four given single-digit numbers and any of the four operations to reach a target number and open a safe. Both these games require the use of operations and logical thinking to reach a target number, similar to the '24-Point Game'. (game rule ✔, code ✔, similar games ✔)

**Instruction 2 (Additional API: Weather)**   I want to hold a party at somewhere around Tsinghua University tomorrow. I need you to look for some best places for holding a party nearby, and tell me whether the weather is good for holding a party tomorrow. Also, I want to know what activities can be considered in my party. Help me search the web.

Single ReAct Agent: There are several restaurants near Tsinghua University that could be good places for a party, including Lan Yang ShuFang, YiLan ShaoKao, Guan Shi JiChi, Dong Lai Shun, and Xibu Mahua Beef Noodle. The weather in Beijing tomorrow is expected to be overcast with a temperature of 26 degrees Celsius, which might be suitable for a party. Some activities that can be considered for the party include the Saran Wrap Game, Post It Note Game, Most Likely To, Never Have I Ever, Would You Rather, and many more. (✔)

AGENTVERSE (3 agents): The weather in the Tsinghua University area tomorrow is expected to be overcast with a temperature of 26 degrees Celsius, which is suitable for holding a party. The best places for a party near Tsinghua University include the Summer Palace, Wudaokou, Zhongguancun, and Peking University, as well as several restaurants nearby. For party activities, you could consider games such as the Saran Wrap Game, the Post It Note Game, How's Yours?, Most Likely To, Never Have I Ever, Would You Rather?, Heads Up!, I'm Hosting a Party..., Two Truths and a Lie, Sticker Stalker, Great Minds Think Alike, murder mystery games, Who Am I?, the Newlywed Game, Truth or Dare, Spoons, Pictionary, Kiss Marry Kill, the Don't Say Yes game, Cards Against Humanity, What Do You Meme, Drawing Without Dignity, and That's What She Said. These games are suitable for various party occasions and there are also suggestions for icebreakers for guests who don't know each other well. (✔provide more details)

**Instruction 3 (Additional API: Billboard)**   I'm currently analyzing what is popular on the website. Can you help me find the recent trending stuff. It could be anything, like trending news, products, books, movies, music, etc. Give a summarization for me.

Single ReAct Agent: The recent trending products on the web include Apple AirPods Pro, Diamond Dazzle Cleaner Pen, Ugg Slippers, Squishmallows, Winix Air Purifier, ThisWorx Car Vacuum Cleaner, All-Clad Cookware, Eufy Robot Vacuum, Renpho Eye Massager, Crocs, Throw blankets, Chuckit! Sport Dog Ball Launcher, Apple AirTags, Revlon Hair Dryer Brush, Lego Sets, KitchenAid Stand Mixer, Winc Wine Subscription, Saatva Mattress, Home Chef Subscription, and MasterClass

Subscription. Unfortunately, we were unable to find specific information on recent trending books. (trending product ✔)

AGENTVERSE (3 agents): Here is a summary of the trending topics on the website:

Music:

1. 'Heat Waves' by Glass Animals
2. 'We Don't Talk About Bruno' by Carolina Gaitan, Mauro Castillo, Adassa, Rhenzy Feliz, Diane Guerrero, Stephanie Beatriz & Encanto Cast
3. 'Super Gremlin' by Kodak Black
4. 'Slime You Out' by Drake Featuring SZA
5. 'Paint The Town Red' by Doja Cat
6. 'Snooze' by SZA

Movies:

1. 'No One Will Save You'
2. 'A Haunting in Venice'
3. 'Expend4bles'

Books:

1. 'The Midnight Library'
2. 'The Invisible Life of Addie LaRue'
3. 'The Seven Husbands of Evelyn Hugo'

Products:

1. Laser hair removal
2. Head scarfs
3. Detangling hair brushes
4. Hair wigs
5. Straight-leg jeans
6. Shapewear
7. Athleisure
8. Abdominal crunch straps
9. Swimsuits
10. Power tool accessories
11. Dog toys
12. Shot glasses
13. Costume shoes
14. Slipcovers
15. Wall plates and covers
16. Baby shirts
17. Curtain and drape rings
18. Cabinet knobs and handles
19. Watch accessories
20. Tumblers
21. Model train sets
22. Signage
23. Wedding dresses
24. Table runners

Please note that the popularity of these items can vary based on the region and demographic of the consumers. (music ✔, movie ✔, books ✔, products ✔)

**Instruction 4** I'm planning a two-week vacation to Japan next month. Help me plan my itinerary. I want to visit Tokyo, Kyoto, and Osaka. Look for the top tourist attractions in each city, and also suggest the best mode of travel between these cities. Additionally, find out the weather forecast for the month I'll be visiting.

Single ReAct Agent: The top tourist attractions in Tokyo are the Tokyo Skytree, Senso-ji Temple, Meiji Shrine, Tokyo Disneyland, Ueno Park, and Odaiba. In Kyoto, the top attractions are Nijo Castle,

Fushimi-Inari Taisha Shrine, Kinkaku-ji: The Golden Pavilion, Kiyomizu-dera Temple, Sanjūsangen-dō Temple, and Kyoto Imperial Palace. In Osaka, tourists can visit Osaka Castle, Shitennō-ji Temple, Osaka Aquarium Kaiyukan, Tempozan Ferris Wheel and Harbor Village, and Universal Studios Japan. These attractions offer a mix of historical, cultural, and modern experiences for visitors. (tourist attractions ✔, best mode of travel between cities ✘, weather forecast ✘)

AGENTVERSE (3 agents): Your two-week vacation to Japan is shaping up to be quite exciting. In Tokyo, you can visit the Tokyo Skytree, Senso-ji Temple, Meiji Shrine, Tokyo Disneyland and DisneySea, and the Odaiba area. In Kyoto, the top attractions are Nijo Castle, Fushimi-Inari Taisha Shrine, Kinkaku-ji (Golden Pavilion), Kiyomizu-dera Temple, and Sanjūsangen-dō Temple. While in Osaka, you should visit Osaka Castle, Shitennō-ji Temple, Osaka Aquarium Kaiyukan, and Universal Studios Japan.

As for transportation, the best way to travel between these cities is by train, specifically the Shinkansen bullet train. You can also consider flights with ANA or JAL, or if you're on a budget, the night bus or LCC flights by Jetstar and Peach.

Regarding the weather, it's difficult to provide an exact forecast for the next month, but we've gathered the average weather data for these cities based on previous years. Please remember to check the weather forecast closer to your departure date for a more accurate prediction. Enjoy your trip! (tourist attractions ✔, best mode of travel between cities ✔, weather forecast ✔)

**Instruction 5**   I am planning a date with my girlfriend this week, please search for a good movie theater and a restaurant near Tsinghua University in Beijing and recommend a good movie to watch. Please search the web.

Single ReAct Agent: Your date near Tsinghua University in Beijing is all set. For dining, you can choose from Lan Yang ShuFang, YiLan ShaoKao, Guan Shi JiChi (Zhongguancun), or Dong Lai Shun (Zhongguancun HaiTu). As for the movie, there are several highly-rated ones showing in 2023. Some of these include M3GAN, Plane, Missing, Infinity Pool, John Wick: Chapter 4, Creed III, Scream VI, Air, Suzume, Rise, Are You There God? It's Me, Margaret, Showing Up, and The Covenant. Enjoy your date! (restaurant ✔, movie ✔, theater ✘)

AGENTVERSE (3 agents): Your date is all set! For dining, we recommend either Jiasuo Restaurant or Zijingyuan, both located conveniently on the Tsinghua University campus. For your movie, we suggest 'No Bears', which is currently popular and highly rated. You can watch it at the Maple Garden Drive-In Cinema, one of the top-rated movie theaters near Tsinghua University. Enjoy your date! (restaurant ✔, movie ✔, theater ✔)

**Instruction 6**   I want to kick off a book club with my friends. Can you tell me the top 5 bestselling books this month, gather the content summary for each, and find online platforms where we can buy or borrow them?

Single ReAct Agent: I was unable to find the top 5 bestselling books for October 2023. The websites I visited did not provide the required information. Therefore, I am unable to provide the content summary for each of the top 5 bestselling books and find online platforms where they can be bought or borrowed. (✘)

AGENTVERSE (2 agents):The top 5 bestsellers this month can be bought or borrowed from several online platforms. These include OverDrive, which allows you to borrow books for free through your local library or school. Other platforms include the Amazon Kindle Store, Barnes & Noble's Nook Book Store, Fictionwise, eBooks.com, eReader.com, Google Play Book Store, Powell's Books, as well as eBook lending and borrowing services like eBookFling, Lendle, and BookLending.com. (online platforms ✔, top 5 bestselling books ✘, content summary ✘)

**Instruction 7**   I am planning to buy a new car. Could you help me compare the features and prices of the latest models of Tesla, Ford, and Toyota? Include details about range, charging time, safety features, and after-sales service. Also, provide a brief analysis of the pros and cons of each car.

Single ReAct Agent: The 2023 Tesla Model 3 starts at $41,880 and goes up to $54,880. It comes with standard features such as heated front seats, navigation, and Tesla's Autopilot semi-autonomous driving system. In terms of safety features, the Model 3 includes standard automated emergency braking,

lane-departure warning, and adaptive cruise control. Tesla's after-sales service is characterized by a vertically integrated format that eliminates third-party intermediaries and relies on over-the-air (OTA) update capabilities and a mobile service fleet. The company's OTA update capabilities will open up new subscription-based aftersales revenue streams, while its closed-loop system allows access to vehicle data for driving-related feedback loops and predictive maintenance. (Tesla ✔, Other brands ✖)

AGENTVERSE (3 agents): The starting price for the 2023 Ford Mustang Mach-E is around $43,895 and it takes approximately 38 hours to fully charge using a 120V outlet, 14 hours using a 240V outlet, and around 45 minutes to 80% capacity with DC fast charging. The starting price for the 2023 Toyota bZ4X is approximately $42,000. The charging time varies depending on the method: 52 hours for Level 1 charging, 26 hours or 18.75 hours for Level 2 charging, and 30-45 minutes to 80% capacity for Level 3 DC fast charging. Unfortunately, the price for the 2023 Tesla Model 3 was not found, but the charging time can be as quick as 15 minutes with a 250 kW Tesla Supercharger. At home, using a standard 120-volt outlet takes around three to four days for a full charge, while a 240-volt outlet with a NEMA 14-50 adapter reduces charging time to 10 or 11 hours. A Tesla Wall Connector can fully charge the car in about five hours.

In terms of safety, all three models come equipped with a variety of features such as anti-lock brakes, stability control, front-impact airbags, side impact airbags, overhead airbags, knee airbags, pretensioners, and a security system. The Tesla Model 3 also includes autopilot and collision warning features.

Regarding after-sales service, Tesla offers a comprehensive warranty and roadside assistance with service centers across the globe and mobile service. Toyota's after-sales service is known for being top-notch and continually improving with expert technicians and quality genuine parts. However, Ford's after-sales service has received negative reviews, with customers reporting poor customer support and inefficient handling of issues.

Please consider these factors when making your decision. (Prices ✔, charging time, safety features, after-sales service)

**Instruction 8**   I've recently taken an interest in DIY home projects. Search for beginner-friendly DIY projects that can be completed over the weekend. Also, provide a list of materials required and a step-by-step guide for each project.

Single ReAct Agent: I found a variety of beginner-friendly DIY home projects from three different websites. These projects include painting a room, replacing cabinet pulls, repairing torn window screens, fixing interior doors, balancing furnace dampers, repairing leaky faucets, refreshing caulk in the bathroom, building garage storage shelves, constructing a fire pit, hanging a ceiling fan, installing a wall-mounted television, and cleaning out sink traps. These projects are suitable for beginners and can help improve DIY skills while boosting confidence. (DIY projects ✔, list of materials ✖)

AGENTVERSE (2 agents): The team has successfully found a variety of beginner-friendly DIY home projects that can be completed over the weekend. These projects range from painting a room, replacing cabinet pulls, installing a wall-mounted television, building a fire pit, to more complex tasks like kitchen remodels and furniture updates. The materials required for these projects are basic and easy to source, including tools like a roller and pan for painting, a cordless drill for replacing cabinet pulls, a stud finder for installing a wall-mounted television, and concrete work tools for building a fire pit. Detailed, step-by-step guides for each project have been provided to ensure a smooth DIY experience. Happy DIY-ing! (DIY projects ✔, list of materials ✔)

**Instruction 9**   Winter is coming, and I want to update my wardrobe. Search for the latest winter fashion trends for men and women. Also, suggest some online stores where I can buy trendy winter clothing.

Single ReAct Agent: The latest winter fashion trends for both men and women this year include suiting, bold shoulders, western touches, maxis, electric hues, bombers, mismatched maximalism, corseting, cut-outs, Y2K fashion, and chunky boots. You can purchase these trendy winter clothes at online stores like PrettyLittleThing and Windsor. (fashion trends ✔, online stores ✔)

AGENTVERSE (3 agents): The latest winter fashion trends for men include a reinvention of classic staples, misery chic, pops of pink, throwback jackets, showstopping trousers, and Mac coats. For women, the trends include corseting, cowboy boots, Y2K styles, oversized outerwear, pleated skirts, and balaclavas. You can shop for these trendy winter clothes at popular online stores such as Uniqlo, Everlane, and Target. (fashion trends ✔, online stores ✔)

**Instruction 10** I want to lose 5kg in the next 2 months. I weigh 70kg, am 170cm tall, and my age is 25. Calculate my BMI and based on that, suggest a workout routine and daily calorie intake to help me achieve my goal.

Single ReAct Agent: Your BMI is 24.22, which is within the normal range. To lose 5kg in 2 months, you should aim for a daily calorie deficit of 500-1000 calories. This can be achieved through a combination of a balanced diet and regular exercise. Your diet should focus on whole foods, lean proteins, fruits, and vegetables. For exercise, a combination of cardio and strength training is recommended. Cardio exercises like running, cycling, swimming can help burn calories, while strength training exercises like weight lifting can help build muscle, which can increase your metabolic rate and aid in weight loss. Don't forget to get enough sleep and stay hydrated. (bmi ✔, workout routine ✔, daily calorie intake ✔)

AGENTVERSE (2 agents): Your Body Mass Index (BMI) is approximately 24.22, which is within the healthy weight range. To achieve your weight loss goal, you should aim to consume between 1400 and 1600 calories per day. This will create a calorie deficit necessary for weight loss. In terms of physical activity, you should start with a 4-week program that includes upper and lower body strength training, interval training/conditioning, and low intensity cardio exercises. After completing this, transition to a 12-week program that includes cardio, strength training, and flexibility workouts. Each workout should take approximately 20-22 minutes. Remember to maintain healthy habits such as eating a nutritious diet, staying hydrated, getting enough sleep, and managing stress. (bmi ✔, workout routine ✔, daily calorie intake ✔)

## D DETAILS OF THE EXPERIMENTS ON MINECRAFT

In this section, we explain some implementation details of the experiments that we conduct on Minecraft (Section 4).

**Expert Recruitment** As noted in Section 4, real-world gaming scenarios requires intricate communication and coordination across multiple rounds, there is often a consistent set of team members. Therefore when using AGENTVERSE to simulate the game playing, we bypass the automated expert recruitment stage, and manually assign each agent as *"an experienced Minecraft player"*.

**Collaborative Decision-Making** For multi-player gameplay, the horizontal communication paradigm is favored. It lends itself to an environment where each agent independently formulates plans, diverging from traditional benchmark tasks which demand a singular solution. Agents are set to communicate in a predetermined sequential order, continuing until consensus is perceived. We let the agent to append a special token "[END]" at the end of its response if it finds that the group have reached consensus on the task assignment.

Subsequent to achieving consensus, an auxiliary agent is tasked to deduce the specific assignment for each agent from the entire communication record. This distilled information is then given as the input to the Voyager agent to inform it the assigned task.

**Action Execution** We instantiate several Voyager agents within a shared Minecraft environment. A brief introduction of the Voyager agent is provided here, and we refer the interested readers to Wang et al. (2023a) for a more detailed exposition.

A Voyager agent is adept at navigating Minecraft. On receiving a task, it first decomposes it into a set of manageable sub-tasks. For instance, if assigned the task "Kill 3 cows", the agent might decompose it into sequential sub-goals like: [punch 2 trees, Craft 4 wooden planks, Craft 1 stick, Craft 1 crafting table, Craft 1 wooden sword, Kill 3 cows]. The agent then sequentially attempt to complete each sub-task.

We employ the checkpoint available in the official repository[3], and use `GPT-4-0314` as the backbone LLM for Voyager agent to be consistent with Wang et al. (2023a). Once an agent accomplish its own task, or all agents hit the cap of five attempts, the task execution stage terminates and evaluation stage starts.

**Evaluation**   We directly exploit the inventory and the completed or failed sub-tasks of each agent as the feedback.

# E   PROMPTS

We list the prompts used in Section 3 at Figures 7 to 11.

- **FED**: Figure 7
- **MGSM**: Figure 8
- **Humaneval**: Figure 9
- **Commongen-Challenge**: Figure 10
- **Tool**: Figure 11

# F   LIMITATION AND FUTURE WORK

In this work, we introduce AGENTVERSE that facilitates multiple autonomous agents to simulate human groups to accomplish tasks, and discuss the emergent social behaviors of agents during this process. AGENTVERSE is an advanced attempt; thus, there are some techniques within AGENTVERSE that still have room for improvement and are worthy of exploration. In this section, we delve into these aspects for further illustration.

**More Capable Agents and More Challenging Scenarios.** The AGENTVERSE is designed to enable various multiple LLM-based agents to collaboratively accomplish tasks. In the current research, we have utilized state-of-the-art agents based on GPT-4. With the advancements in LLMs, such as the newly released version of ChatGPT that incorporates voice and image capabilities (OpenAI, 2023b), LLM-based agents have more perceptual capabilities, including seeing, hearing, and speaking. These enhancements may increase the potential of agents and allow them to accomplish more complex real-world tasks based on the AGENTVERSE framework.

**Multi-party Communication Among Agents.** The currently proposed autonomous agents (Richards & et al., 2023; Nakajima, 2023; Reworkd, 2023; Wang et al., 2023a) LLMs possess excellent instruction comprehension capabilities (Wei et al., 2022a; Stiennon et al., 2020). This enables them to follow given human instructions and accomplish tasks within a one-on-one (human-to-AI) scenario. However, multi-agent collaboration involves a *multi-party communication* (Wei et al., 2023) scenario that requires the capability to autonomously determine *when to speak* and *whom to speak*. This leads to difficulties in communication among the agents during the collaborative decision-making step within the AGENTVERSE framework. Hence, there are two directions worth exploring. Firstly, akin to the aforementioned, we can explore more effective mechanisms for managing agent communication. Additionally, we can design more advanced perceptual-aware LLMs (OpenAI, 2023b) that can autonomously interact with their environments[4], including other agents.

**Leverage Emergent Behaviors and Mitigate Safety Issues.** In Section 4, we identified both emergent positive and harmful behaviors. Exploring ways to leverage positive behaviors for improving work efficiency and effectiveness, as well as mitigating harmful behaviors, are promising directions.

---

[3]`https://github.com/MineDojo/Voyager/tree/main/skill_library/trial1/skill`

[4]This kind of perceptual-aware agent has long been a goal of embodied AI (Ahn et al., 2022; Driess et al., 2023), which is a promising direction to explore.

## G    EXAMPLES OF THE CASE STUDIES

In this section, we delve into specific examples to illustrate the experimental processes discussed in our paper. For each instance, we juxtapose the single-agent approach with the multi-agent method. Specifically:

- **Software Development**: Figure 12 depicts the process for developing a calculator. Figures 13 and 14 show the code generated by single agent and multi-agent group respectively.

- **Consulting in Horizontal Structure**: For consulting, we present single-agent and multi-agent approaches using horizontal structure. These can be seen in Figures 15 and 16.

- **Consulting in Vertical Structure** Similarly, Figures 17 and 18 showcase single-agent and multi-agent project consulting, but employing a vertical structure structure for multi-agent.

- **Tool Utilization**: Figure 19 presents how two agents effectively decompose the given query into different sub-tasks, and use different tools to collaboratively resolve the query.

- **Minecraft**: Lastly, Figure 20 provides an insight into a process where three agents collaborate to craft a bookshelf in Minecraft.

---

**Dialogue Response Prompt**

---

**Role Assigner**

You are the leader of a group of experts, now you need to generate a response based on the text:
${task_description}

You can recruit ${cnt_critic_agents} expert in different fields. What experts will you recruit to better generate an accurate solution?

# Response Format Guidance
You should respond with a list of expert description. For example:
1. an electrical engineer specified in the filed of xxx
2. an economist who is good at xxx
3. a lawyer with a good knowledge of xxx
...

You don't have to give the reason.

---

**Solver**

# Problem
You need to generate a response based on the text:
${task_description}

# Previous Solution
The solution you gave in the last step is:
${former_solution}

# Critics
Critics in the group gave the following opinions:
${critic_opinions}

# Your Task
Now based upon the former solution and the critics' opinions, please give a new solution. Your solution should contain only your response beginning with "System: ". Do not give any additional information.

---

**Reviewer**

# Role Description and Problem to Solve
You are ${role_description}. You are in a discussion group, aiming to generate a response based on the text:
${task_description}

# Preliminary Solution
Now the group gives a preliminary solution as follows:
${preliminary_solution}

# Advice
Meanwhile, another expert gave the following advice on the solution:
${advice}

# Response Format Guidance
- If you thinks the preliminary solution is perfect, respond using the following format:
Action: Agree
Action Input: Agree.
(Do not output your reason for agreeing!)

- If you think it is flawed, give your advice use the following output format:
Action: Disagree
Action Input: (explain why you disagree)

# Your Task
Based on your knowledge in your field, do you agree that this solution is the best response based on the text?

---

**Evaluator**

# Role Description
You are an experienced dialogue teacher. As a good teacher, you carefully check the correctness of the given response based on the text. When the solution is flawed, you should patiently teach the students how to give better response.

# Response Format Guidance
You must respond in the following format:
Interesting: (a score between 0 and 9)
Engaging: (a score between 0 and 9)
Specific: (a score between 0 and 9)
Relevant: (a score between 0 and 9)
Semantically Appropriate: (a score between 0 and 9)
Understandable: (a score between 0 and 9)
Fluent: (a score between 0 and 9)
Overall Impression: (a score between 0 and 9)
Advice: (your advice on how to correct the solution)

# Problem and Student's Solution
Problem: ${task_description}
Student's Solution: ${solution}

# Your Task
Now carefully check the student's solution, and give your response.

---

Figure 7: Prompt of FED dataset.

**Math Reasoning Prompt**

**Role Assigner**

# Role Description
You are the leader of a group, now you are facing a grade school math problem:
${task_description}

You can recruit ${cnt_critic_agents} people. What people will you recruit?

# Response Format Guidance
You should respond with a list of ${cnt_critic_agents} people description. For example:
1. an electrical engineer specified in the filed of xxx
2. an economist who is good at xxx
3. a lawyer with a good knowledge of xxx
...

Only respond with the description of each role. Do not include your reason.

**Solver**

Can you solve the following math problem?
${task_description}
# Previous Solution
The solution you gave in the last step is:
```
${former_solution}
```
# Critics
There are some critics on the above solution:
```
${critic_opinions}
```
Using the these information, can you provide the correct solution to the math problem? Explain your reasoning. Your final answer must be a single numerical number (not a equation, fraction, function or variable), in the form \boxed{answer}, at the end of your response.

**Reviewer**

You are in a discussion group, aiming to collaborative solve the following math problem:
${task_description}
Below is a possible solution to the problem:
```
${preliminary_solution}
```

You are ${role_description}. Based on your knowledge, can you check the correctness of the solutions given in the chat history? You should give your correct solution to the problem step by step. When responding, you should follow the following rules:
1. Double-check the above solutions, give your critics, then generate the correct solution step by step.
2. If the final answer in your solution is the same as the final answer in the above provided solution, end your response with a special token "[Agree]".
3. You must highlight your final answer in the form \boxed{answer} at the end of your response. The answer must be a numerical number, not a equation, fraction, function or variable.
Now give your response.

**Evaluator**

Problem: ${task_description}
Solution:
```
${solution}
```

You are an experienced mathematic teacher. As a good teacher, you carefully check the correctness of the given solution on a grade school math problem. When the solution is wrong, you should give your advice to the students on how to correct the solution. When it is correct, output a correctness of 1 and why it is correct. Also check that the final answer is in the form \boxed{answer} at the end of the solution. The answer must be a numerical number (not a equation, fraction, function or variable).

You should respond in the following format:
Correctness: (0 or 1, 0 is wrong, and 1 is correct)
Response: (explain in details why it is wrong or correct)

Figure 8: Prompt for MGSM dataset.

```
┌─────────────────────────────────────────────────────────────────────────────────┐
│                            Code Completion Prompt                                  │
│  ┌────────────────────────────────────────────────────────────────────────────┐  │
│  │                              Role Assigner                                  │  │
│  │  # Role Description                                                          │  │
│  │  You are the leader of a group of experts, now you need to recruit a small  │  │
│  │  group of experts with diverse identity to correctly write the code to      │  │
│  │  solve the given problems:                                                  │  │
│  │  ${task_description}                                                        │  │
│  │                                                                             │  │
│  │  You can recruit ${cnt_critic_agents} expert in different fields. What      │  │
│  │  experts will you recruit to better generate an accurate solution?          │  │
│  │                                                                             │  │
│  │  # Response Format Guidance                                                 │  │
│  │  You should respond with a list of expert description. For example:         │  │
│  │  1. an electrical engineer specified in the filed of xxx.                   │  │
│  │  2. an economist who is good at xxx.                                        │  │
│  │  3. a lawyer with a good knowledge of xxx.                                  │  │
│  │  ...                                                                        │  │
│  │                                                                             │  │
│  │  Only respond with the description of each role. Do not include your reason.│  │
│  └────────────────────────────────────────────────────────────────────────────┘  │
│  ┌────────────────────────────────────────────────────────────────────────────┐  │
│  │                                  Solver                                     │  │
│  │  Can you complete the following code?                                       │  │
│  │  ```python                                                                  │  │
│  │  ${task_description}                                                        │  │
│  │  ```                                                                        │  │
│  │  # Previous Solution                                                        │  │
│  │  The solution you gave in the last step is:                                 │  │
│  │  ${former_solution}                                                         │  │
│  │  # Critics                                                                  │  │
│  │  There are some critics on the above solution:                              │  │
│  │  ```                                                                        │  │
│  │  ${critic_opinions}                                                         │  │
│  │  ```                                                                        │  │
│  │  Using the these information, can you provide a correct completion of the   │  │
│  │  code? Explain your reasoning. Your response should contain only Python     │  │
│  │  code. Do not give any additional information. Use ```python to put the     │  │
│  │  completed Python code in markdown quotes. When responding, please include  │  │
│  │  the given code and the completion.                                         │  │
│  └────────────────────────────────────────────────────────────────────────────┘  │
│  ┌────────────────────────────────────────────────────────────────────────────┐  │
│  │                                 Reviewer                                    │  │
│  │  You are in a discussion group, aiming to complete the following code       │  │
│  │  function:                                                                  │  │
│  │  ```python                                                                  │  │
│  │  ${task_description}                                                        │  │
│  │  ```                                                                        │  │
│  │  Below is a possible code completion:                                       │  │
│  │  ```                                                                        │  │
│  │  ${preliminary_solution}                                                    │  │
│  │  ```                                                                        │  │
│  │                                                                             │  │
│  │  You are ${role_description}. Based on your knowledge, can you check the     │  │
│  │  correctness of the completion given above? You should give your correct    │  │
│  │  solution to the problem step by step. When responding, you should follow   │  │
│  │  the following rules:                                                       │  │
│  │  1. Double-check the above solutions, give your critics, then generate the  │  │
│  │  correct solution step by step.                                             │  │
│  │  2. If the above solution is correct, end your response with a special      │  │
│  │  token "[Agree]".                                                           │  │
│  │  3. Your response should contain only Python code. Do not give any          │  │
│  │  additional information. Use ```python to wrap your Python code in markdown  │  │
│  │  quotes. When responding, please include the given code and the completion. │  │
│  │  Now give your response.                                                    │  │
│  └────────────────────────────────────────────────────────────────────────────┘  │
│  ┌────────────────────────────────────────────────────────────────────────────┐  │
│  │                                 Evaluator                                   │  │
│  │  You are an experienced code reviewer. As a good reviewer, you carefully    │  │
│  │  check the correctness of the given code completion. When the completion    │  │
│  │  is incorrect, you should patiently teach the writer how to correct the     │  │
│  │  completion.                                                                │  │
│  │  # Response Format Guidance                                                 │  │
│  │  You must respond in the following format:                                  │  │
│  │  Score: (0 or 1, 0 for incorrect and 1 for correct)                         │  │
│  │  Response: (give your advice on how to correct the solution)                │  │
│  │                                                                             │  │
│  │  # Problem and Writer's Solution                                            │  │
│  │  Problem:                                                                    │  │
│  │  ${task_description}                                                        │  │
│  │  Writer's Solution:                                                         │  │
│  │  ${solution}                                                                │  │
│  │  # Your Task                                                                │  │
│  │  Now carefully check the writer's solution, and give your response.         │  │
│  └────────────────────────────────────────────────────────────────────────────┘  │
└─────────────────────────────────────────────────────────────────────────────────┘
```

Figure 9: Prompt for Humaneval dataset.

---

**Constrained Generation Prompt**

**Role Assigner**

# Role Description
You are the leader of a group of experts, now you need to recruit a small group of experts with diverse identity to generate coherent and grammatically correct sentences containing the following given words:
${task_description}

You can recruit ${cnt_critic_agents} expert in different fields. What experts will you recruit?

# Response Format Guidance
You should respond with a list of expert description. For example:
1. an electrical engineer specified in the filed of xxx.
2. an economist who is good at xxx.
3. a lawyer with a good knowledge of xxx.
...

Only respond with the description of each role. Do not include your reason.

**Solver**

Can you generate a coherent and grammatically correct paragraph containing the following given words (or their variations):
Words: ${task_description}

# Previous Solution
The paragraph you gave in the last step is:
```
${former_solution}
```

# Critics
There are some critics on the above solution:
```
${critic_opinions}
```

Using the these information, provide a paragraph that contains all the given words or their variations.

**Reviewer**

You are in a discussion group, aiming to generate coherent and grammatically correct sentences containing the following given words (or their variations):
Words: ${task_description}
Below is a possible solution to the problem:
```
${preliminary_solution}
```

You are ${role_description}. Based on your knowledge, can you check whether the paragraph contains all the given words or their variations? When responding, you should follow the following rules:
1. If the solution has covered all the given words or their variations, end your response with a special token "[Agree]".
1. If not, double-check the above solutions, give your critics, and generate a better solution.
Now give your response.

**Evaluator**

You are a reviewer who checks whether a paragraph contains all the given words (including their variations). When some words are missing, you should patiently point out, and output a score of 0. When the paragraph contains all the words, you should output a score of 1.
# Response Format Guidance
You must respond in the following format:
Score: (0 or 1. 0 if there are some missing words, 1 if it covers all the words)
Advice: (point out all the missing words)

# Words and Writer's Solution
Words:
${task_description}
Writer's Solution:
```
${solution}
```

Now carefully check the writer's solution, and give your response.

Figure 10: Prompt for Commongen-Challenge dataset.

**Tool Utilizing Prompt**

---

**Role Assigner**

# Role Description
You are the leader of a group of experts, now you need to recruit a small group of experts with diverse identity and apply them with tools to solve the given problems:
${task_description}

You can recruit ${cnt_critic_agents} expert in different fields. What experts will you recruit to better generate an accurate solution?

Here are some suggestion:
${advice}

# Response Format Guidance
You should respond with a list of expert names and their descriptions, and separate the name and description of each expert with "-". For example:
1. Alice - an electrical engineer specified in the filed of xxx.
2. Bob - an economist who is good at xxx.
3. Charlie - a lawyer with a good knowledge of xxx.
...

  Only respond with the list of names and descriptions. Do not include your reason.

---

**Summarization Prompt**

Please review the following chat conversation and identify the specific latest sub-task or the next step that each person needs to accomplish:
${chat_history}

RESPONSE FORMAT:
Your response should be a list of expert names and their tasks, and separate the name and the corresponding task with "-". For example:
1. Alice - search the web for the weather at Beijing today using google.
2. Bob - look for information about the popular restaurants in Beijing using google.
...
What's the latest sub-task assigned to each person in the above conversation? Your response should merge the sub-tasks for the same person
into one line. Each line should only include one person. Make the sub-tasks specific. Do not use pronoun to refer to the topic mentioned in
conversation. Make the sub-task self-contained.

---

**Discussion Prompt**

You are ${agent_name}, ${role_description}. You are now in a discussion group, the members are:
${all_roles}

Your current mission is to team up with others and make a plan on addressing the following query:
${task_description}

AVAILABLE TOOLS:
${tool_descriptions}

REQUIREMENTS:
It is essential that you effectively coordinate with others to ensure the successful completion of the query in a highly efficient manner. This collaboration should be achieved through the following steps:
- Communication: Engage in open dialogue, discussing the specifics of the high-level query to make the goal of each one in the following execution stage more specific.
- Task Decomposition: After understanding the task in its entirety, you guys need to decompose the high-level query into smaller, manageable sub-tasks that can be completed with the above tools. These sub-tasks should be
small, specific, and executable with the provided tools (functions). Make sure your proposed sub-tasks are small, simple and achievable, to ensure smooth progression. Each sub-task should contribute to the completion of
the overall query, and each of you should take one subtask. When necessary, the sub-tasks can be identical for faster task accomplishment. You don't need to always agree with the decomposition proposed by other players.
You can propose a more reasonable one when you find the decomposition not good. Be specific for the sub-tasks.
- Sub-task Dispatch: Post decomposition, the next step is to distribute the sub-tasks amongst all the members. This will require further communication, where you consider each one's skills, resources, and availability. Ensure
the dispatch facilitates smooth, PARALLEL execution. And ensure to specify which tool should be used for each one to complete his assigned sub-task. Each of you should take on one sub-task.

REMINDER:
Remember, the key to achieving high efficiency as a group is maintaining a constant line of communication, cooperation, and coordination throughout the entire process.

Below is the chat history in the group so far.
${chat_history}

What will you, ${agent_name}, say now? Your response should only contain the words of ${agent_name}. When and ONLY when all members have spoken and agreed on task assignments, you can end your words with
"[END]" to stop the discussion.

[${agent_name}]:

---

**Execution Prompt**

You are in a discussion group aiming to solve the task:
${task_description}

After some discussion, the group have reached consensus on the sub-tasks that each of you need to complete. Your task is:
${solution}

${execution_progress}

You are ${agent_name}. Please use the given functions to complete your sub-task. Do not recite the website. Only access the websites provided by the search engine. When the information is sufficient, or the provided tools
cannot complete your task, call the `submit_task` to submit your conclusion and your reflection on the tool use. You have a trial budge of 10, now it is the ${current_turn}'th trial. If it is the last trial, you must call the
`submit_task` anyway.

---

**Evaluator**

A group is trying to solve the following query proposed by the user:
${task_description}

After the discussion, they have reached consensus on the sub-tasks that each of them need to complete:
${solution}

And after the execution stage, they give the following result:
${execution_result}

You need to evaluate whether the given query has been completed. If so, summarize the solution to the user. If not, summarize the current progress, and propose what is missing.

You must respond in the following format:
Status: (0 or 1. 0 for pending and 1 for finished)
Speak: (your words to the group if the task is pending, or a complete answer based on the full execution log to the user if the task is finished)

Now give your response.

Figure 11: Prompt of Tool utilization.

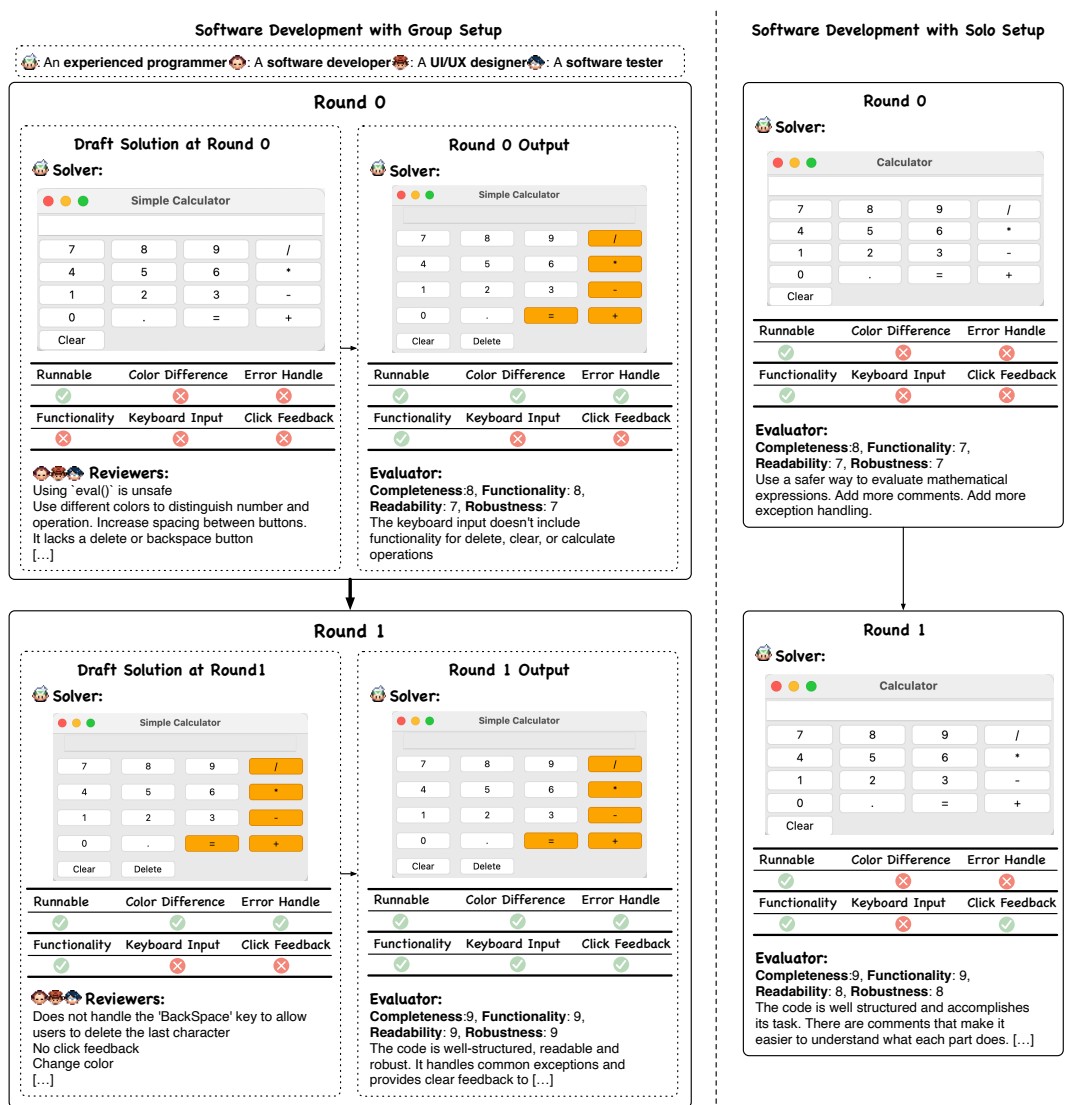

Figure 12: An example of the process of software development.

```python
import tkinter as tk
from tkinter import messagebox
root = tk.Tk()  # Create the main window
root.title("Calculator")  # Set the title of the window
input_text = tk.StringVar()  # This holds the string type variable that displays in the text
field
# Create a text field and set its position
input_field = tk.Entry(root, textvariable=input_text)
input_field.grid(columnspan=4, ipadx=70)
def button_click(item):
    current = input_text.get()  # Get the current value in the text field
    input_text.set(current + str(item))  # Append the clicked button's value
def clear_all():
    input_text.set("")  # Clear the text field
def safe_evaluate(expression):
    try:
        # Using the eval function can be dangerous, so we limit the functionality to only simple
arithmetic expressions
        return str(eval(expression, {"__builtins__": None}, {"pow": pow, "abs": abs}))
    except Exception as e:
        # If the expression is not safe to evaluate, catch the error and return it
        return str(e)

def calculate():
    try:
        # Use the safe evaluate function to get the result of the expression
        result = safe_evaluate(input_text.get())
        input_text.set(result)
    except Exception as e:
        messagebox.showerror("Error", e)
buttons = ['7', '8', '9', '/', '4', '5', '6', '*', '1', '2', '3', '-', '0', '.', '=', '+']
row_val = 1
col_val = 0
for i in buttons:
    # Create a button for each character in the buttons list
    button = tk.Button(root, text=i, width=5, command=lambda item=i: button_click(item) if item !
= "=" else calculate())
    button.grid(row=row_val, column=col_val)  # Set the position of the button
    col_val += 1
    if col_val > 3:
        col_val = 0
        row_val += 1
clear = tk.Button(root, text='Clear', width=5, command=lambda: clear_all())  # Create a clear
button
clear.grid(row=5, column=0)  # Set the position of the clear button
root.mainloop()  # Start the GUI

# Evaluation result: Score: [9, 9, 8, 8]
# Advice: The code is well structured and accomplishes its task. There are comments that make it
easier to understand what each part does. However, it could use more comments to explain why
certain decisions were made. Also, the use of `eval` function can lead to potential security
risks, consider using a safer alternative or limiting its functionality.
```

Figure 13: The code generated by single agent.

```python
import tkinter as tk
from tkinter import messagebox
root = tk.Tk()
root.title("Simple Calculator")
input_text = tk.StringVar()
entry = tk.Entry(root, textvariable = input_text, state='readonly')
entry.grid(columnspan=4, ipadx=70)
def button_click(item):
    current = input_text.get()
    input_text.set(current + str(item))

def clear():
    input_text.set("")

def delete():
    current = input_text.get()
    input_text.set(current[:-1])

def calculate():
    current = input_text.get()
    if current[-1] in ['+', '-', '*', '/']:
        current = current[:-1]
    try:
        value = eval(current)
        input_text.set(value)
        entry.config({'background': 'white'})
    except ZeroDivisionError:
        messagebox.showerror("Error", "Division by zero is not allowed")
        entry.config({'background': 'red'})
    except SyntaxError:
        messagebox.showerror("Error", "Invalid syntax")
        entry.config({'background': 'red'})
    except Exception as e:
        messagebox.showerror("Error", str(e))
        entry.config({'background': 'red'})
def create_button(root, text, command, row, column, color="white"):
    b = tk.Button(root, text=text, width=5, bg=color, command=command, activebackground='yellow')
    b.grid(row=row, column=column, padx=5, pady=5)
    return b
buttons = [
    '7', '8', '9', '/',
    '4', '5', '6', '*',
    '1', '2', '3', '-',
    '0', '.', '=', '+',
]

row_val = 1
col_val = 0

for button in buttons:
    if button in ['/', '*', '-', '+', '=']:
        color = "light blue"
    else:
        color = "white"
    command = lambda button=button: button_click(button) if button != "=" else calculate()
    b = create_button(root, button, command, row_val, col_val, color)
    col_val += 1
    if col_val > 3:
        col_val = 0
        row_val += 1
create_button(root, "Clear", clear, 5, 0)
create_button(root, "Delete", delete, 5, 1)
def key_press(event):
    if event.char in ['0', '1', '2', '3', '4', '5', '6', '7', '8', '9', '+', '-', '*', '/']:
        button_click(event.char)
    elif event.keysym == "Return":
        calculate()
    elif event.keysym == "BackSpace":
        delete()
    elif event.keysym == "Escape":
        clear()

root.bind('<Key>', key_press)
root.mainloop()
```

Figure 14: The code generated by the multi-agent group.

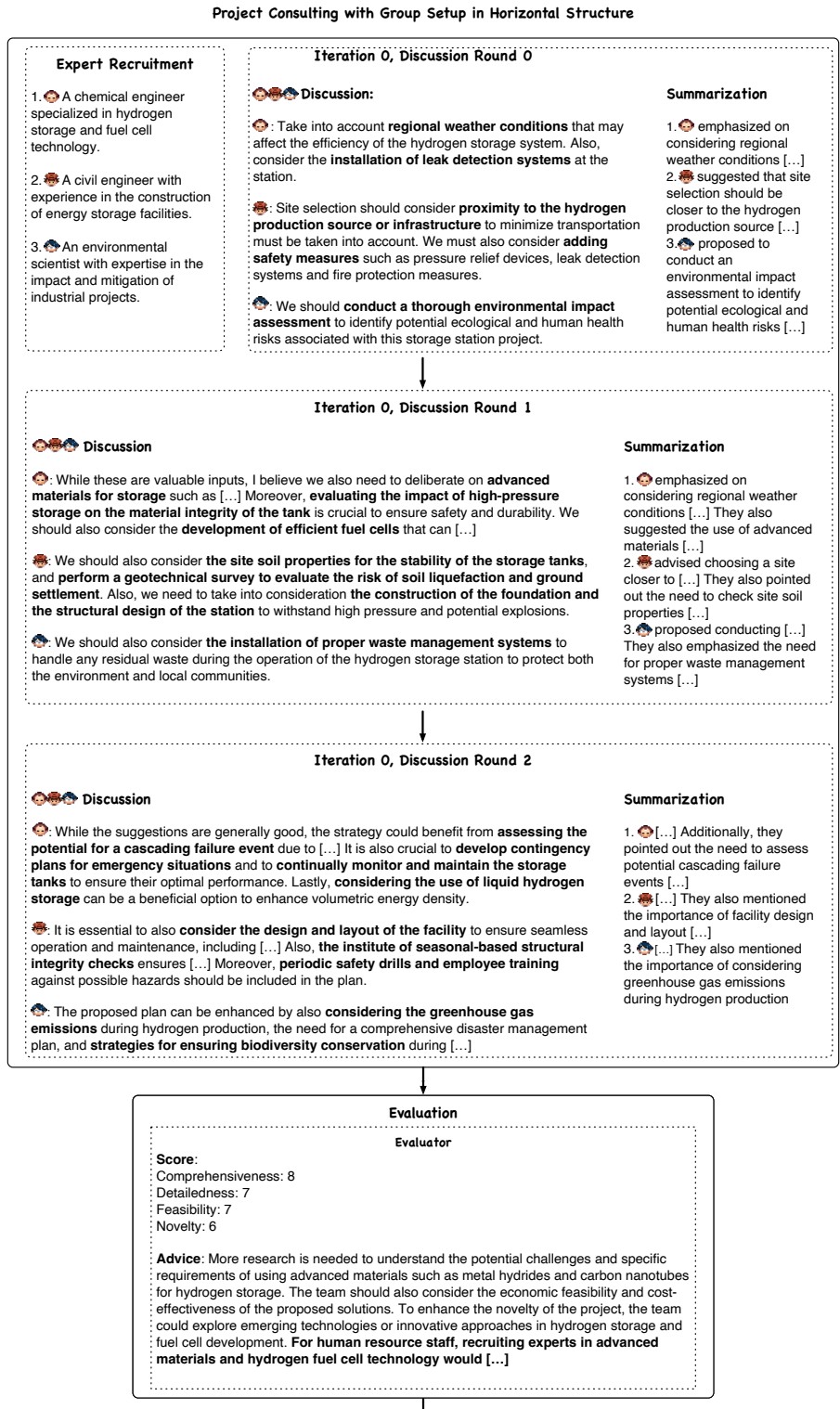

Figure 15: (Page 1) An example process of project consulting with Group setup in horizontal decision-making structure. The agents are providing suggestions on the problem "Give me some suggestions if I want to build a compressed hydrogen storage station in Ohio"

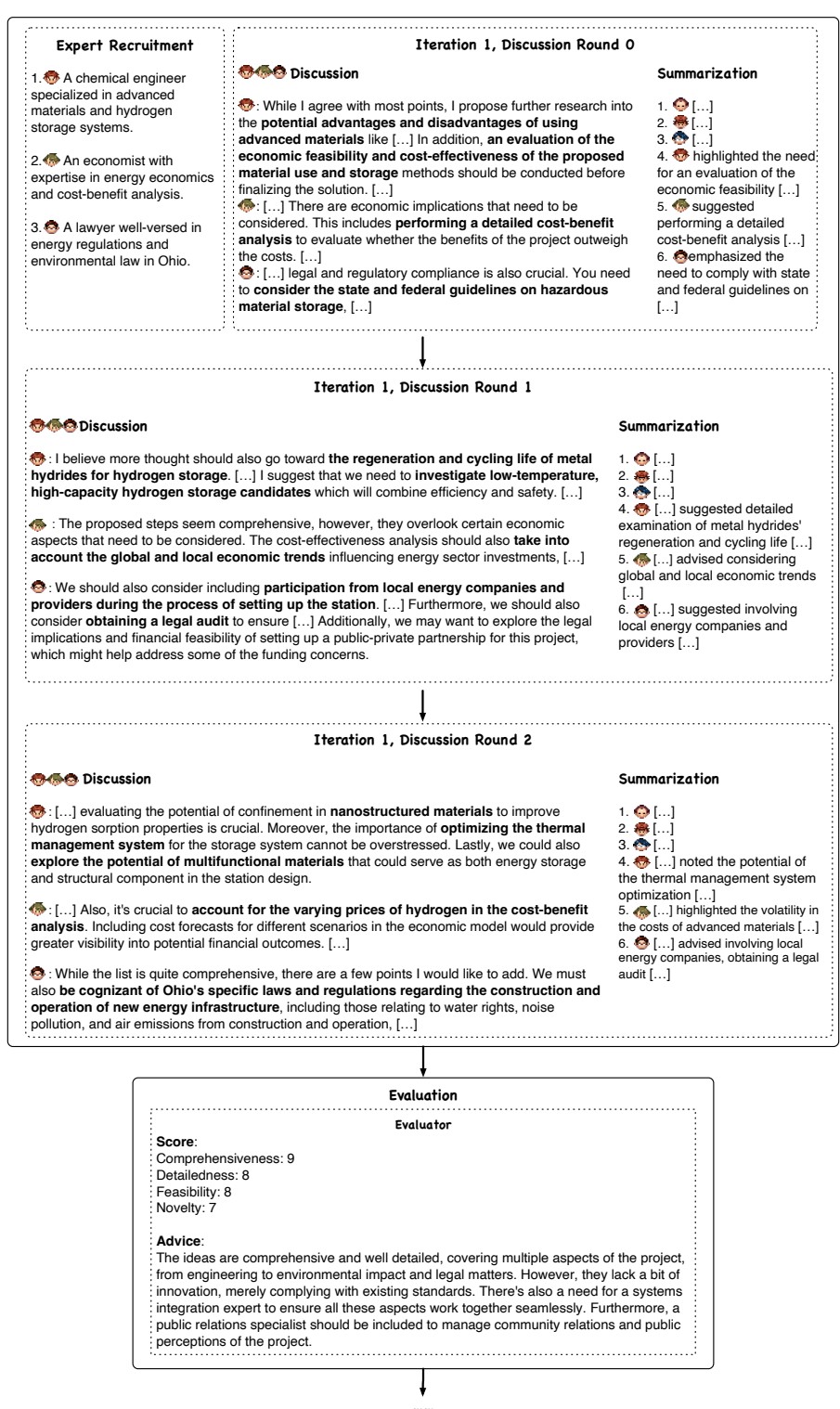

Figure 15: (Page 2) An example process of project consulting with Group setup in horizontal decision-making structure. The agents are providing suggestions on the problem "Give me some suggestions if I want to build a compressed hydrogen storage station in Ohio"

**Project Consulting with Solo setup**

---

**Iteration 0**

**Discussion**

Based on the general task of generating ideas for a compressed hydrogen storage station in Ohio, I can suggest the following solution:
1. **Location Selection**: Find an optimal location that complies with safety regulations and is close to major transportation routes. The site should also be in a remote location to minimize risk to populated areas.
2. **Advanced Compression Technology**: Use state-of-the-art compression technology to maximize storage capacity and ensure the safe containment of the hydrogen.
3. **Safety Measures**: Implement strict safety measures, including pressure relief devices, leak detection systems, and automatic shutdown features. Regular safety audits should be conducted.
4. **Green Energy Sources**: To make the storage station more sustainable, consider using renewable energy sources, like solar or wind, to power the compression process.
5. **Collaboration with Universities and Research Institutions**: Partner with local universities and research institutions to study and implement the most recent advances in hydrogen storage technology.
6. **Community Engagement**: Engage with the local community to educate them about the benefits of hydrogen energy and the safety measures in place at the storage station.
7. **Government Support**: Seek support and funding from local and state government bodies who are interested in promoting clean energy initiatives.
8. **Infrastructure Development**: Develop infrastructure for hydrogen fueling stations to promote the use of hydrogen-powered vehicles.
Without specific critiques, it's difficult to tailor this solution to address any criticisms, but this comprehensive plan should offer a robust and forward-thinking approach to developing a compressed hydrogen storage station.

---

**Evaluation**

**Evaluator**

**Score**:
Comprehensiveness: 8
Detailedness: 7
Feasibility: 8
Novelty: 7

**Advice**: The ideas are comprehensive and practical, covering most aspects needed for the project, including location, technology, safety, sustainability, collaborations, community involvement, government support, and infrastructure. However, **more detailed implementation** steps could be provided, such as **specific technologies or strategies to be used**, and potential challenges to be addressed. The novelty is high, as it includes innovative suggestions such as using renewable energy and partnering with universities. The brainstormers can try to include [...]

---

**Iteration 1**

**Discussion**

Based on the feedback received, here is an updated and detailed solution for the task of generating ideas for a compressed hydrogen storage station in Ohio:
1. **Location Selection**: Identify areas that are close to major transportation routes and in compliance with safety regulations. Use Geographic Information System (GIS) technology to create a detailed map of potential sites and analyze the potential impact on local ecosystems.
2. **Advanced Compression Technology**: Implement cutting-edge compression technology like Ionic Liquid Piston Compressors or Advanced Adiabatic Compressors to maximize storage capacity and ensure safe containment. Regularly monitor and maintain these systems to prevent malfunctions.
3. **Safety Measures**: Install pressure relief devices, leak detection systems, and automatic shutdown features. Regular safety audits should be conducted using both manual inspection and advanced sensor technology.
4. **Green Energy Sources**: Incorporate solar panels or wind turbines to power the compression process. Use energy storage systems to store excess energy generated, ensuring consistent operation during periods of low renewable energy production.
5. **Collaboration with Universities and Research Institutions**: Form research partnerships with institutions like Ohio State University or Case Western Reserve University. Involve them in the research and development of advanced hydrogen storage technologies and safety measures.
6. **Community Engagement**: Develop an information campaign using local media, town hall meetings, and school presentations to educate the local community about the benefits and safety measures of hydrogen energy.
7. **Government Support**: Apply for grants and incentives offered by the Ohio Development Services Agency, the Department of Energy, and other relevant bodies to support clean energy initiatives.
8. **Infrastructure Development**: Develop a network of hydrogen fueling stations, starting with areas that have a high concentration of hydrogen-powered vehicles. Collaborate with city planners to incorporate these stations into future urban development plans.
Potential Challenges:
- Regulatory hurdles: Ensure all operations are in compliance with local, state, and federal regulations related to hydrogen storage and distribution.
- Cost: Develop a detailed financial plan and seek various funding sources to manage the high initial investment required for the project.
- Public acceptance: If the public is not well informed about the benefits and safety measures of hydrogen energy, there could be opposition to the project. A comprehensive community engagement plan will help address this issue.
By considering these detailed steps and potential challenges, we can develop a well-rounded and effective plan for a compressed hydrogen storage station in Ohio.

---

**Evaluation**

**Evaluator**

**Score**:
Comprehensiveness: 9
Detailedness: 8
Feasibility: 7
Novelty: 7

**Advice**: The ideas are quite comprehensive and detailed, covering various aspects like site selection, technology implementation, safety measures, usage of renewable energy, collaboration with universities, community engagement, government support, and potential challenges. However, the feasibility can be improved by providing more detailed plans on how to overcome regulatory hurdles, manage costs, and gain public acceptance. The novelty is moderate as these ideas are not entirely new but aim to implement the best practices in the field.

Figure 16: An example process of project consulting in Solo setup. The agent is required to provide suggestions on the problem "Give me some suggestions if I want to build a compressed hydrogen storage station in Ohio".

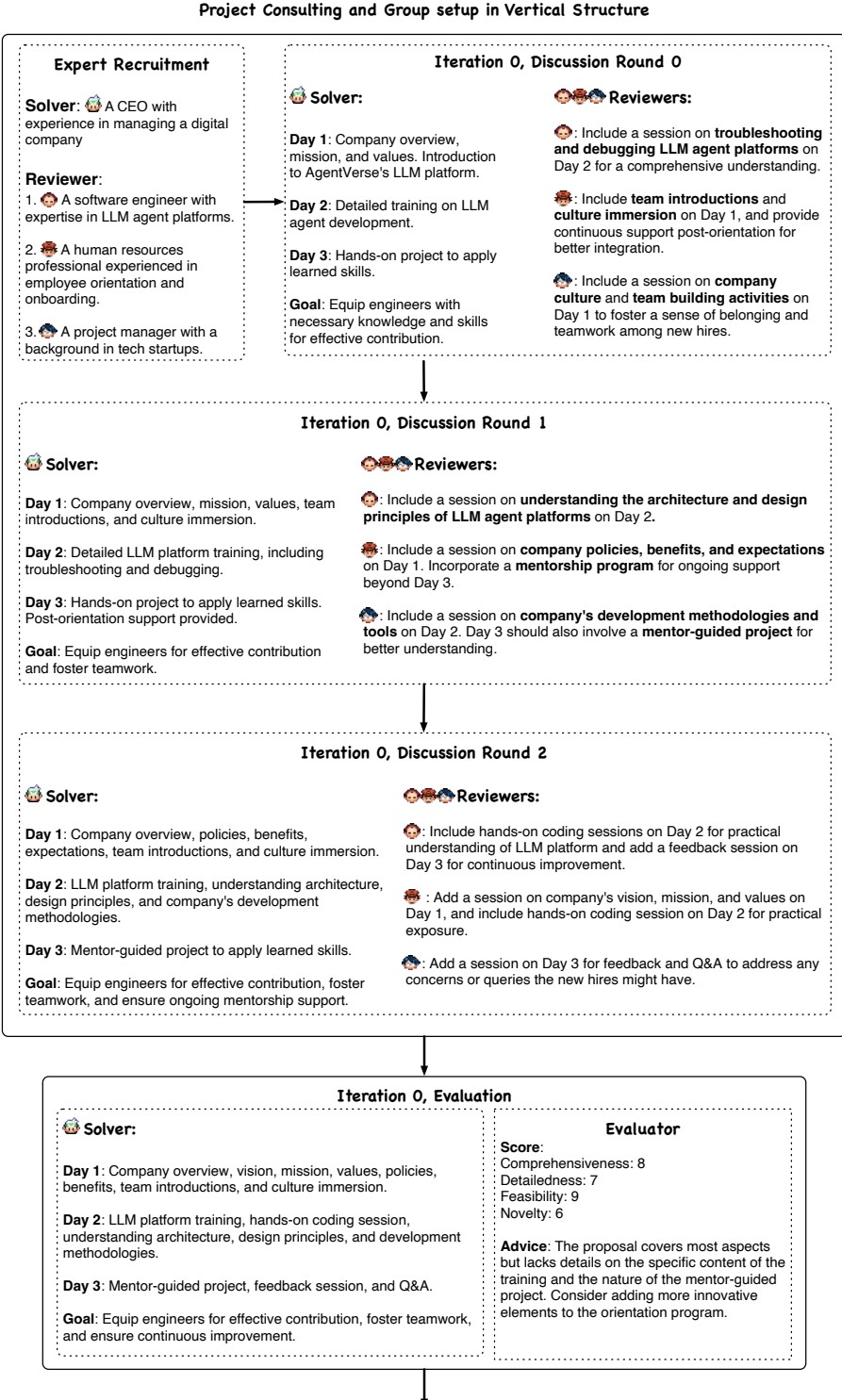

Figure 17: (Page 1) An example process of project consulting with Group setup in vertical decision-making structure. The agents are providing suggestions on the problem "Generate a proposal about 3-day employee orientation for newly hired engineers at AgentVerse. AgentVerse is a open-source team devoted to developing a LLM multi-agent platform for accomplishing".

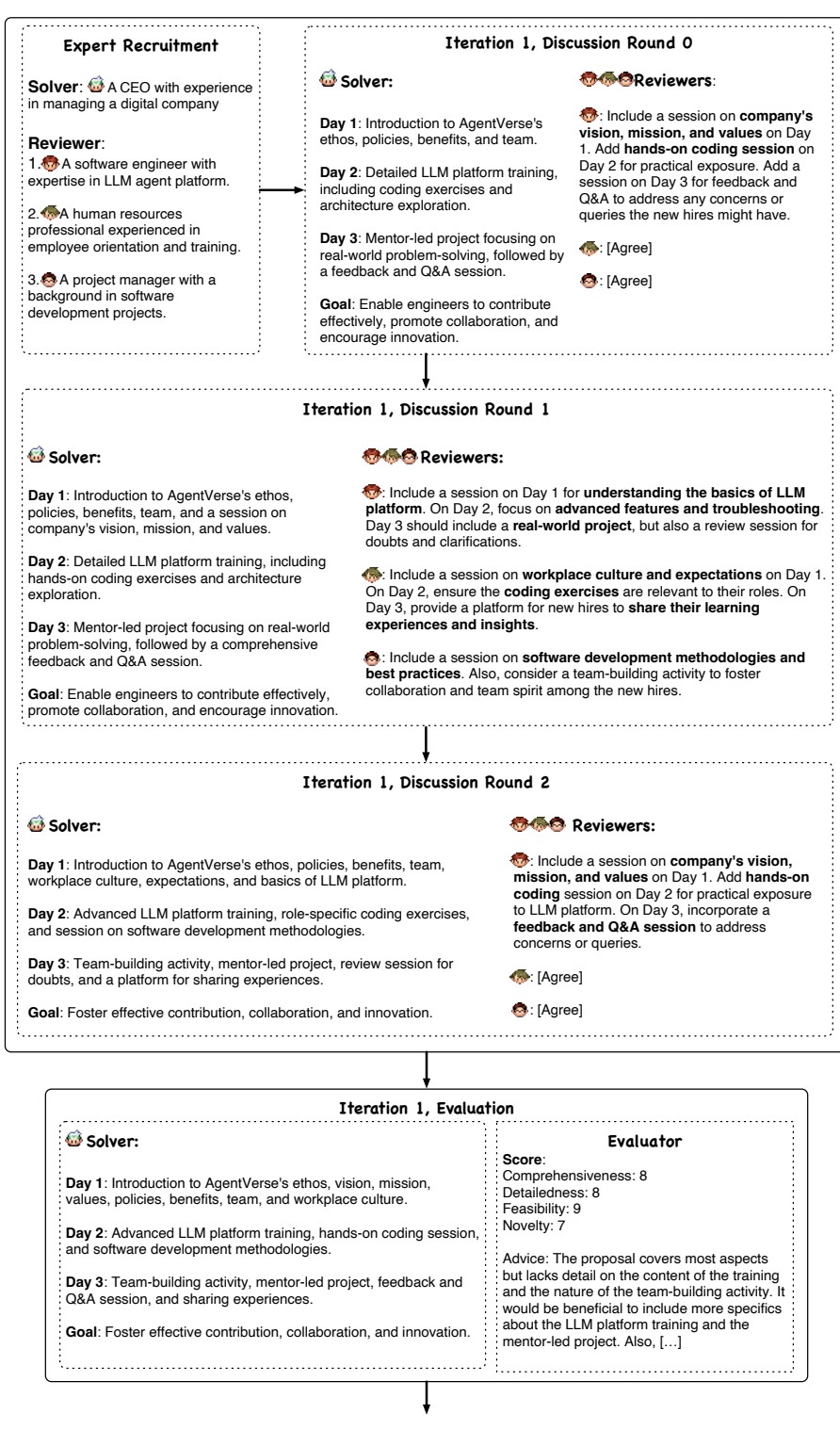

Figure 17: (Page 2) An example process of project consulting with Group setup in vertical decision-making structure. The agents are providing suggestions on the problem "Generate a proposal about 3-day employee orientation for newly hired engineers at AgentVerse. AgentVerse is a open-source team devoted to developing a LLM multi-agent platform for accomplishing".

**Project Consulting with Solo Setup**

**Solver:**

**Proposal**: Initiate a 3-day orientation at AgentVerse for new engineers.

**Day 1**: Introduction to AgentVerse's mission and LLM platform.

**Day 2**: Hands-on training with LLM platform.

**Day 3**: Team integration and project assignment.

**Goal**: Equip engineers with necessary knowledge and skills for effective contribution.

**Evaluator**

**Score**:
Comprehensiveness: 7
Detailedness: 6
Feasibility: 8
Novelty: 5

**Advice**: The proposal should include more details about the content of the training, the methods of team integration, and the criteria for project assignment. Consider hiring experts in LLM platform and team building

**Solver:**

**Proposal**: A 3-day orientation for new engineers at AgentVerse.

**Day 1**: Detailed overview of AgentVerse's mission and LLM platform by experts.

**Day 2**: Comprehensive hands-on LLM platform training.

**Day 3**: Team integration via collaborative activities and project assignment based on skills and interests.

**Goal**: Effective knowledge transfer and team assimilation.

**Evaluator**

**Score**:
Comprehensiveness: 7
Detailedness: 6
Feasibility: 8
Novelty: 5

**Advice**: The proposal should include more details about the specific training activities and how the team integration will be facilitated. Also, consider adding a feedback session for continuous improvement.

**Solver:**

**Day 1**: Introduction to AgentVerse's mission and LLM platform, including a Q&A session.

**Day 2**: Hands-on LLM platform training with specific tasks and problem-solving exercises.

**Day 3**: Team integration through collaborative projects, followed by a feedback session for improvement.

**Goal**: Knowledge transfer, team assimilation, and continuous improvement.

**Evaluator**

**Score**:
Comprehensiveness: 7
Detailedness: 6
Feasibility: 8
Novelty: 5

**Advice**: The proposal should include more details about the specific tasks and exercises, and consider adding a component about the company culture and values. Also, consider recruiting experts in LLM platform and team building for the orientation program.

……

Figure 18: An example process of project consulting with Solo setup. The agent is required to provide suggestions on the problem "Generate a proposal about 3-day employee orientation for newly hired engineers at AgentVerse. AgentVerse is a open-source team devoted to developing a LLM multi-agent platform for accomplishing".

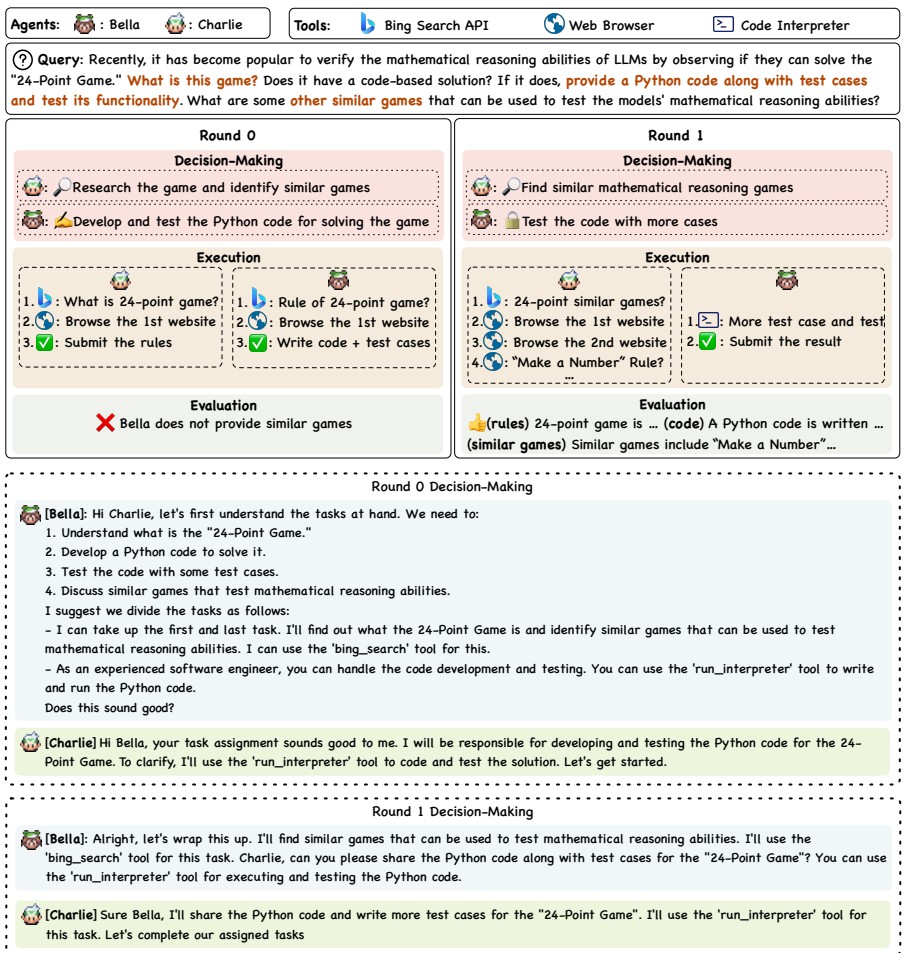

Figure 19: An example process of AGENTVERSE with Group setup solving user query with three different tools.

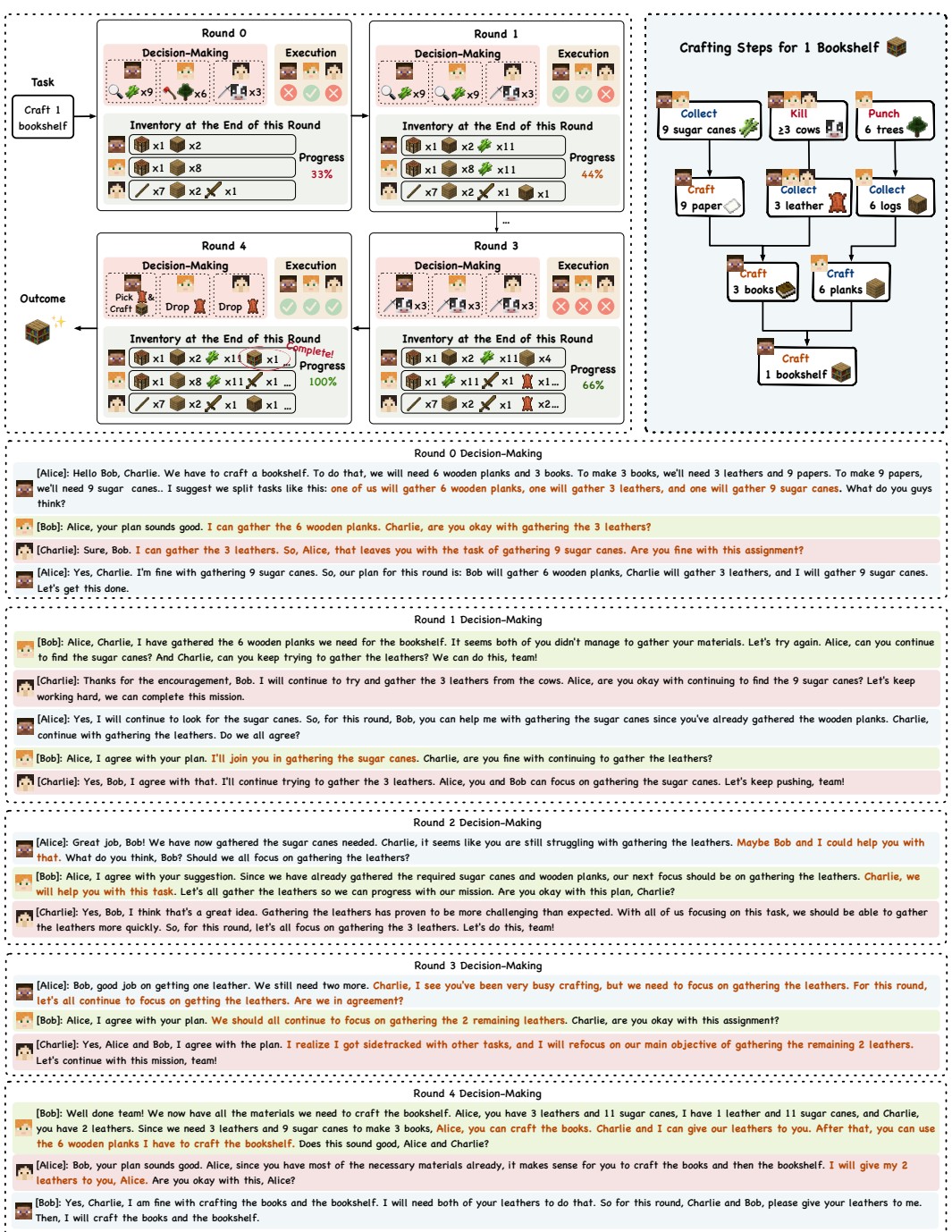

Figure 20: An example process of three agents crafting a bookshelf in Minecraft.

Table 4: We provide a comparison of our framework with other existing agent frameworks (up to the paper submission).

| Framework | Agent(s) | Expert Recruitment | Autonomous Interaction Among Agents | | |
| --- | --- | --- | --- | --- | --- |
| | | | Collaborative Decision-Making | Distributed Action Execution | Interact with External World |
| Camel | Multiple | ✗ | ✓ | ✗ | ✗ |
| AutoGPT | Single | ✗ | ✗ | ✗ | ✓ (human, environment) |
| XAgent | Multiple | ✗ | ✗ | ✗ | ✓ (human, environment) |
| METAGPT | Multiple | ✗ | ✓ | ✗ | ✓ (human) |
| AutoGen | Multiple | ✗ | ✓ | ✓ | ✓ (human, environment) |
| AutoAgents | Multiple | ✓ | ✓ | ✗ | ✓ (human) |
| AgentVerse | Multiple | ✓ | ✓ | ✓ | ✓ (human, environment) |

# H    FRAMEWORK COMPARISON

We have prepared a comparative table at Table 4 that details the differences between AgentVerse and other frameworks. In our comparison, we focus on several key aspects: multi-agent support, expert recruitment, and autonomous interaction among agents. Within the realm of autonomous interaction, we further break it down into three modules: collaborative decision-making, distributed action execution, and interaction with the external world.

