# OpenReview forum: "AgentVerse: Facilitating Multi-Agent Collaboration and Exploring Emergent Behaviors"
_ICLR.cc/2024/Conference — ICLR 2024 poster_

### Official Review · Reviewer_Mwtr · 2023-11-01

**Soundness:** 3 good
**Presentation:** 3 good
**Contribution:** 2 fair
**Rating:** 6
**Confidence:** 3

**Summary:**

This method introduces AgentVerse, a multi-agent framework powered by LLMs to solve problems from the perspectives of several experts (as simulated by a prompted LLM agent). In AgentVerse, the user enters in a command/question and then the framework will collect responses to the command/question from the LLM after its been prompted to act as a particular type of expert. Then, the expert agents will collaborate and attempt to synthesize a concise response for the user, either in an unordered/parallel manner or ordered/serial manner. The result of this decomposition produces better responses (as measured by various problem solving benchmarks) that are agnostic to the underlying LLM (GPT3.5-Turbo and GPT4).

**Strengths:**

This paper is a novel and interesting usage of LLMs to solve problems. By prompting to answer the users inputs from the perspective of different experts, the LLMs can produce more nuanced understanding of a problem. This provides more insight into a question as well as more interpretability for why the LLM is responding in the way that it is. This finding is simple and common sense but novel, to my knowledge.

**Weaknesses:**

The experimentation on the AgentVerse framework is promising but quantitative results are seriously lacking in this paper. The case studies provide interesting insight into how the AgentVerse framework deals with problems but is not yet significant enough. For instance, Table 1 and Table 2 provide the only quantitative results in the paper, including the appendix.

I also am curious about the necessity of framing this as a multi-agent problem. Can sufficient prompting of the agent provide enough context that multiple agents are not necessary (beyond the solo and CoT baselines)?

**Questions:**

1. Do the authors have any additional quantitative results, with included uncertainty, to support their claims beyond Table 1 and 2?

---

> ### Author Response · Authors · 2023-11-15
>
> Thank you for your comments on our experiments. We've identified three major concerns from your reviews and will address each of them.
>
> ---
>
> **Q1: Lacking quantitative experiments.**
>
> A1: We appreciate your feedback regarding the quantitative experiments of our research. Our paper includes five benchmark datasets, a number we consider substantial for evaluating multi-agent systems. These benchmarks were chosen to be diverse and evaluate different aspects of our framework.
>
> Furthermore, the emergent behaviors and communication patterns we have shown in the case studies are non-trivial. **These elements, though less tangible than numerical benchmark results, are central to advancing the understanding and capabilities of multi-agent systems**. The case studies in our work are designed to mirror real-world challenges more closely than conventional benchmark datasets and, therefore, **provide valuable insights into the practical applicability of AgentVerse**. We kindly request the consideration of these qualitative aspects as equally, if not more, important than quantitative experiments. The nuanced understanding they provide into agent interaction dynamics is vital for the advancement of multi-agent systems.
>
> ---
>
> **Q2: The necessity of framing the problem as a multi-agent one.**
>
> A2: We believe that while sophisticated prompting can enhance single-agent capabilities, the multi-agent framework offers distinct advantages:
>
> 1. **Efficiency of Multi-Agent System:** An individual agent, even with sophisticated prompting, may face limitations in efficiency when tackling complex tasks. Multi-agent systems excel in this aspect, especially in real-world scenarios involving intricate multi-tool utilization (Section 3.3) or collaborative goals in gaming environments (Section 4). By coordinating and collaborating, multiple agents can operate more efficiently than a single agent in these contexts since they have a clear division of labor, and less distracting information in their memory.
> 2. **Complementarity of Single and Multi-Agent Approaches:** Advanced prompting techniques can indeed create more capable agents. This does not contradict with multi-agent system approach. Multi-agent system can also benefit from more capable agents. This synergy allows for leveraging the strengths of single-agent efficiency and multi-agent collaboration.
> 3. **Exploring Multi-Agent Dynamics:** Our research delves into some of the intricate properties of multi-agent communication, shedding light on how heterogeneous agents can collaborate effectively. This exploration reveals the potential of multi-agent systems, a domain where single-agent frameworks fall short. The dynamics within multi-agent interactions are not just an extension of single-agent capabilities but constitute a distinct and valuable area of research. Our work represents a significant step in demonstrating the potential and worth of multi-agent studies based on LLMs.
>
> ---
>
> **Q3: Experimental results with uncertainty**
>
> A3: We additionally run the experiments of GPT-3.5-turbo for 3 times, and report the averaged performance and the standard deviation on each dataset:
> |                        | CoT             | Solo            | Group           |
> | ---------------------- | --------------- | --------------- | --------------- |
> | Conversation           | $83.7_{\pm1.0}$ | $84.0_{\pm1.7}$ | $86.8_{\pm1.2}$ |
> | Creative Writing       | $79.5_{\pm0.2}$ | $92.4_{\pm0.2}$ | $90.2_{\pm0.4}$ |
> | Mathematical Reasoning | $81.2_{\pm0.6}$ | $83.2_{\pm1.5}$ | $82.8_{\pm0}$   |
> | Coding                 | $72.4_{\pm0.3}$ | $73.4_{\pm1.9}$ | $74.8_{\pm0.7}$ |

---

> > ### Author Response · Authors · 2023-11-21
> >
> > We would like to express our sincere gratitude for your valuable insights and constructive feedback, which have been instrumental in enhancing the quality of our paper. In response to your comments, we have diligently **conducted additional experiments and made significant efforts to address your concerns, as outlined in our detailed response**.
> >
> > As the discussion phase is nearing its conclusion, with just two days remaining, we are keenly awaiting your thoughts and feedback post-rebuttal. Your further input would be highly beneficial to us.
> >
> > Should you require any more clarifications, or if there are additional experiments that might help in evaluating our work more thoroughly, please do not hesitate to let us know. We are more than willing to engage in further discussions to alleviate any remaining concerns.
> >
> > Lastly, if you find that our revisions and responses have satisfactorily addressed your initial comments, we would be grateful if you would consider supporting the paper by increasing the score. Your support is invaluable to us.
> >
> > Thank you once again for your time and effort in reviewing our work.

---

> ### Author Response · Authors · 2023-11-23
>
> Dear reviewer Mwtr,
>
> We understand the demands on your time and appreciate any effort you can make to review our response. Your continued feedback is crucial in guiding our work to meet the conference’s standards fully. We are eager to hear your thoughts on the improvements made and hope our response align closely with your expectations.
>
> We also humbly request you to revisit our response to all the reviewers' comments and consider the possibility of raising the score you have kindly attributed to our work. Your expertise and fair judgment are crucial in this process, and any additional feedback or guidance you could provide would be immensely appreciated.

---

### Official Review · Reviewer_4SVa · 2023-11-01

**Soundness:** 2 fair
**Presentation:** 3 good
**Contribution:** 2 fair
**Rating:** 6
**Confidence:** 2

**Summary:**

In this paper, the authors proposed a framework named Agentverse for automatically synthesizing collaborative specialized agents. The framework simulates the problem-solving procedures of human groups, and allows for dynamic adjustment of group members based on current progress. Finally, some experiments were conducted to show the effectiveness of their work.

**Strengths:**

1.	This paper involves a framework for promoting collaboration among multiple agents in problem-solving;
2.	The contributions of the paper is relevant for the LLM-based multiagent systems;
3.	The results of this paper is interesting and significant in multiple autonomous agents empowered by LLMs.

**Weaknesses:**

1.	The motivation of this paper seems unclear. It does not clearly show the necessity of AGENTVERSE compared to existing Agent models.
2.	There is no comparison between the framework proposed in this paper and the existing LLM-based agent framework should be listed, such as whether it generates agent dynamically, or the number of agents is limited, etc. Multiple dimensions of indicators should be investigated in detail to explain the advantages of your proposed framework.
3.	In related work section, it does not summarize and classify the related work of LLM-based automotive agents since these two years. LLMs have been widely used as core controllers for automotive agents that can comply with specific objectives, such as Auto-GPT and MetaGPT, etc.
4.	While the paper mentions the release of the AGENTVERSE codebase, it does not provide more specific information about how and where readers can access it. It would be beneficial to provide clear repository details or a direct link.

**Questions:**

1.	For collaborative decision-making, two typical communication structures i.e., horizontal and vertical structures are considered. More details are necessary. For example, in the horizontal structure, the integration function f can be a majority voting like summarization mechanism?
2.	When the number of tasks is 10, AGENTVERSE is superior to single ReAct agent in the experiment of tool utilization capabilities, how about the performance when the number of tasks increases or the tasks are heterogeneous?
3.	What does the data with respect to coding capabilities in Table 2 mean?

---

> ### Author Response · Authors · 2023-11-15
>
> We thank you for your constructive feedback. Below, we address your concerns and questions raised in the review:
>
> ---
>
> **Q1: Motivation of our paper.**
>
> A1: The evolution of AI research is progressively shifting from the development of individual agents to fostering their ability to collaborate effectively. Our research is driven by a vision to **explore and enhance this collaborative potential among AI agents**.
>
> The core of our motivation lies in addressing the complexity of **enabling multiple agents to communicate, coordinate, and collaborate efficiently**. The current LLMs shaped by techniques such as SFT and RLHF are tailored to function as 'assistants,' primarily responding to human queries. However, there is a distinct gap in their ability to act as 'agents' that autonomously interact with each other and navigate diverse environments, whether virtual or physical.
>
> To bridge this gap, it's crucial to design and implement a nuanced communication mechanism that facilitates effective interaction among LLM agents. In response to this need, we have developed AgentVerse. Our framework not only demonstrates effectiveness in various benchmark datasets but also showcases its applicability in a range of **real-world scenarios**. Moreover, our observations of **emergent behaviors** within AgentVerse underscore the untapped potential of LLM agents in multi-agent systems, paving the way for groundbreaking advancements in AI collaboration.
>
> ---
>
> **Q2: Comparison with other frameworks**
>
> A2: We have prepared a comparative table detailing how AgentVerse and other frameworks differ.
>
> Comparison Table:
>
> - Multiple Agent
> - Expert Recruitment
> - Autonomous Interaction Among Agents
>     - Collaborative Decision-Making
>     - Distributed Action Execution
>     - Interact with external world
>
> |  |  |  | autonomous interaction | autonomous interaction | autonomous interaction |
> | --- | --- | --- | --- | --- | --- |
> |  | Single/Multiple Agent | Expert Recruitment | Collaborative Decision-Making (Multi-agent Communication) | Distributed Action Execution | Interact with external world  |
> | Camel | Multiple | x | ✓ | x | x |
> | AutoGPT | Single | x | x | x | ✓ (human, environment) |
> | XAgent | Multiple | x | x | x | ✓ (human, environment) |
> | METAGPT | Multiple | x | ✓ | x | ✓ (human) |
> | AutoGen | Multiple | x | ✓ | ✓ | ✓ (human, environment) |
> | AutoAgents | Multiple | ✓ | ✓ | x | ✓ (human) |
> | AgentVerse | Multiple | ✓ | ✓ | ✓ | ✓ (human, environment) |
>
> ---
>
> **Q3: Missing related work**
>
> A3: Thank you for pointing out the omission of recent developments in LLM-based automotive agents in our related work section. We will add a paragraph discussing the related work on autonomous agents with notable examples like Auto-GPT, MetaGPT, and others that have emerged as key players in the field.
>
> ---
>
> **Q4: AgentVerse codebase**
>
> A4: We have uploaded an anonymized version of our codebase, you can refer to [https://anonymous.4open.science/r/AgentVerse-EFC23/](https://anonymous.4open.science/r/AgentVerse-EFC23/README.md). In this code, we further extend AgentVerse to multi-agent simulation. You can refer to the task solving part of our code.
>
> ---
>
> **Q5: Details of collaborative decision-making**
>
> A5: For the question on the integration function f in the horizontal structure, it can be summarization or ensemble, as we have said in Section 2.2. When it is summarization, it summarizes the discussion among the agents; when it is ensemble, we simply concatenate all the messages from all the agents as the output. We acknowledge that some details may still be vague, and additional explanations on the two communication structures will be added.
>
> ---
>
> **Q6: Tool utilization performance with more and heterogeneous tasks**
>
> A6: We acknowledge the existence of datasets like ToolBench (https://arxiv.org/abs/2307.16789) for complex multi-tool tasks. However, their lack of a robust evaluation method, primarily relying on LLMs, limits their reliability. Our preliminary experiments with ToolBench also faced technical challenges: most of the APIs they provide have failed. The problem is still not perfectly solved even after contacting the authors. Furthermore, other available tool datasets such as APIBench (https://arxiv.org/abs/2305.15334), either only involve single tool or are overly simplistic for our multi-agent tool utilization context.
>
> We invite you to review Appendix B, which compares the performances of a single ReAct agent and AgentVerse. This comparison clearly demonstrates AgentVerse's superior performance over the single-agent approach. We did not do any cherry picking. Based on these findings, we are confident in AgentVerse’s ability to outperform single-agent setups in complex tool utilization scenarios.
>
> ---
>
> **Q7: Table 2 meaning**
>
> A7: It shows the Pass@1 of GPT-3.5-turbo and GPT-4 on Humaneval dataset. For both models, multi-agent setting consistently outperforms solo and CoT settings.

---

> > ### Author Response · Authors · 2023-11-21
> >
> > We deeply appreciate the time and effort you have invested in reviewing our paper, and your insightful feedback has been a key factor in enhancing the overall quality of our work. Following your suggestions, we have **made the motivation for our paper much clearer, added comparisons with other frameworks, and tried our best to address all other concerns**.
> >
> > As the discussion period is approaching its conclusion, we are eagerly anticipating your thoughts on our revision. Your additional feedback at this stage would be invaluable to us.
> >
> > Please let us know if there are any further clarifications or specific experiments you would recommend to strengthen our paper. We are committed to engaging in a productive dialogue to address any residual issues.
> >
> > Moreover, if our revisions and detailed responses meet the concerns initially raised, we would be grateful if you could consider supporting the paper by increasing the score. Your acknowledgment of our efforts in improving the paper would be immensely appreciated.
> >
> > Thank you once again for your invaluable contributions to refining our research.

---

> > ### Comment · Reviewer_4SVa · 2023-11-23
> >
> > I thank the authors for their detailed feedback. They have addressed some of my concerns, so I decide to increase the score.

---

> ### Author Response · Authors · 2023-11-23
>
> Dear reviewer 4SVa,
>
> We are eager to hear your thoughts on the improvements made and hope our revisions align more closely with your expectations. Also, we are pleased to update you that our responses to other reviewers have been positively received, with a majority of the reviewers now favoring acceptance. Your valuable feedback has been integral to this progress. As we approach the final stages of evaluation, your insights remain crucial. We kindly request your review of the latest version at your earliest convenience. Your expertise is greatly appreciated and will be instrumental in determining the final decision on our work. We look forward to your response with great anticipation.

---

### Official Review · Reviewer_rnqE · 2023-11-05

**Soundness:** 3 good
**Presentation:** 4 excellent
**Contribution:** 4 excellent
**Rating:** 6
**Confidence:** 3

**Summary:**

The authors present AGENTVERSE, a multi-agent framework designed to bring together the collective abilities of specialized agents, aiming to surpass the capabilities of individual agents when tackling complex tasks. The framework operates through a structured process inspired by human collaborative problem-solving, following a sequence of analysis/action steps:  Expert Recruitment, Collaborative Decision-Making, Action Execution, and Evaluation. The authors' experiments demonstrate that AGENTVERSE outperforms individual agents in tasks requiring general understanding, reasoning, and coding, especially when state-of-the-art models are employed. The authors analyze behavioral patterns emerging from the AGENTVERSE process, observing emergent behaviors in agent groups, such as volunteering, conformity behavior, and destructive behavior. The experiments and analysis of the paper suggest that orchestrating a collaborative approach among autonomous agents can lead to improved problem-solving effectiveness and the development of new collaborative behaviors.

**Strengths:**

Originality:  This work matures and extends prior concepts for using collaborative agents, and their analysis suggests ways in which emergent behavior might be considered in such systems.

Quality:  This is a well composed paper, outlining and analyzing a nice idea.  The experiments appropriately demonstrate the effectiveness of the approach, and the discussion of results are good.

Clarity:  The overall concept is well presented, the effectiveness adequately captured by the experiments, and non-quantitative behavioral aspects are both well framed in prior work and clearly articulated -- both when the method demonstrates superior performance, and when solo agents outperform the group.

Significance:  This paper advances the state of understanding of collaborative agents in problem solving, and is a good contribution to the field.

**Weaknesses:**

1. Although not a significant weakness, performance drivers in quantitative experiments could use additional elaboration, vs. the terse tabular summary.

2. Complimentary and similar approaches are emerging rapidly, and some quantitatively outperform this method.  I think this paper presents a distinct approach, and the results might be enhanced by contemplating the integration of other methods.  As just one example*, a very recent preprint using different methods (https://arxiv.org/abs/2310.04406) demonstrates similar or better performance improvements on the code dataset.  A discussion of how to merge other ideas into this approach might be helpful.

*I have no affiliation with the authors of the referenced work, and have no knowledge of that work beyond having recently run across it on arXiv.

**Questions:**

1. Can you comment on the extensibility of your approach, for instance in the potential future work section?

---

> ### Author Response · Authors · 2023-11-15
>
> We are grateful for the suggestion to include more detailed explanation on experimental results and more discussion on the recent work. Here’s how we revise our paper accordingly:
>
> ---
>
> **Q1: Possible integration with recent work**
>
> A1: We acknowledge the value of incorporating recent advancements into AgentVerse. We think that many of these studies, focused on enhancing individual agents’ reasoning, complement our work. Importantly, they can be seamlessly integrated into AgentVerse. For example, the method you mentioned (https://arxiv.org/abs/2310.04406) is a tree-of-thought-like method that significantly enhances single-agent reasoning. The improved reasoning abilities of individual agents, when combined with our framework's collaborative dynamics, may lead to a more robust and effective problem-solving system. We view such advancements as complementary, enabling potential synergistic improvements to our system, rather than as alternative approaches.
>
> ---
>
> **Q2: Future work**
>
> A2: As the capability of single agent evolves, we believe that exploring how to enable these agents to move beyond working in isolation will become an important research topic. AgentVerse, with its modular design, facilitates experimentation with various agent teams and tasks. In our paper's Appendix E, we discuss potential extensions, and here we would like to discuss how AgentVerse can be extended to this research.
>
> 1. **More Capable Agents and More Challenging Scenarios.** We can easily incorporate more capable agents from recent research, and apply them to challenging scenarios, such as more sophisticated embodied agent scenarios. Our experiments in the Minecraft environment, a sandbox game, demonstrate its applicability for agents in virtual world. This success suggests that extending AgentVerse to other embodied simulations, or even real-world physical scenarios, is not only feasible but a promising avenue for future exploration. Minecraft serves as a proof of concept, showing how AgentVerse's principles can be adapted to environments where agents interact in more physically grounded settings.
> 2. **Enhanced Communication.** There are several topics that can be further explored in multi-agent communication. For example, the communication structure. Though we only design two straightforward and intuitive communication structures, i.e., horizontal and vertical, further exploration can be made based on our code and experiments by customizing and replacing the module. Also, the research on communication language, protocol, etc. can all be considered within AgentVerse.
> 3. **Leverage Emergent Behaviors and Mitigate Safety Issues.** The emergent behaviors observed in our tool utilization and Minecraft scenarios provide valuable insights. How can we harness the beneficial emergent behaviors will be a research topic for multi-agent research. Also, the harmful behaviors we observed also indicates the necessity to study the agent safety within multi-agent context. It may be a new scenario that needs to be considered in alignment.
>
> ---
>
> **Q3: Elaboration on performance drivers.**
>
> A3: Our framework's performance is driven by several key factors:
>
> 1. **Role Assignment**: Assigning expert identities to agents enhances accuracy, as supported by related work (e.g., https://arxiv.org/abs/2305.14930) and evidenced in our consulting case (Figure 2). We've also added an experiment highlighting this, showing improved results in GPT-3.5-Turbo CoT tasks with role assignment. We run each setting for 3 runs and report the averaged result and the standard deviation:
>
> |  | w/o role | w/ role |
> | --- | --- | --- |
> | Math Reasoning | $72.4_{\pm 0.3}$ | $73.7_{\pm 0.5}$ |
> | Coding | $81.2_{\pm 0.6}$ | $82.4_{\pm 0.7}$ |
>
> 2. **Environmental Feedback**: Utilizing environmental feedback, like coverage tests in Commongen and unit tests in Humaneval, significantly boosts performance. This is explained in Appendix A, illustrating the impact on datasets where such feedback is applicable.
> 3. **LLM Communication Capabilities**: As noted in Section 3.1, the ability of LLMs like GPT-3.5-turbo and GPT-4 to handle contradictions during decision-making stages is crucial. GPT-4's superior handling of contradictions indicates the importance of robust communication capabilities for maximizing the benefits of a multi-agent system.

---

> > ### Comment · Reviewer_rnqE · 2023-11-18
> >
> > I thank the authors for their thoughtful reply and update to the paper.   I believe this is a paper worth publication, and retain my original rating of 6.

---

> ### Author Response · Authors · 2023-11-23
>
> Dear reviewer rnqE,
>
> We sincerely appreciate your recognition of the updates and improvements made in our response, and we are grateful for your positive remarks regarding its suitability for publication.
>
> We also wish to draw your attention to the extensive efforts we have invested in addressing the concerns raised by the other reviewers. These efforts have not only improved specific aspects of our work but also contributed to a more comprehensive and robust paper overall.
>
> Given the additional experimental results, the clearer comparison with other frameworks, and other improvements made in response to all reviewers' feedback, we humbly request you to consider raising the score you have assigned. We think that these collective enhancements might merit a revised evaluation.
>
> We fully respect your expertise and judgment in this matter and understand the importance of maintaining the integrity of the review process. Any additional insights or suggestions you may offer will be highly appreciated and carefully considered.
>
> Thank you once again for your time and thoughtful consideration of our work.

---

### Official Review · Reviewer_Zbb4 · 2023-11-10

**Soundness:** 3 good
**Presentation:** 3 good
**Contribution:** 3 good
**Rating:** 6
**Confidence:** 5

**Summary:**

The paper introduces AgentVerse, a multi-agent framework that facilitates collaboration and explores emergent behaviors. The framework is designed to orchestrate a group of expert agents to accomplish tasks more efficiently and effectively than a single agent. The paper presents experiments that demonstrate the effectiveness of AgentVerse in various domains, including text understanding, reasoning, coding, tool utilization, and embodied AI. The contributions of the paper include the development of AgentVerse, the demonstration of its effectiveness in various domains, and the exploration of emergent behaviors that arise from collaboration among agents.

**Strengths:**

- The paper introduces AgentVerse, an effective framework for promoting collaboration among multiple agents in problem-solving. The process of expert recruitment, collaborative decision-making, action execution, and evaluation is very interesting.
- The authors have conducted abundant experiments over different kinds of applications, including text understanding, reasoning, coding, tool utilization, and embodied AI to show the efficacy of their proposed framework.
- The paper has discussed some interesting emergent behaviors during the interactions between agents.
- The paper is generally well-written and is easy to follow. The visualization can help people understand the framework and the analysis.

**Weaknesses:**

- Evaluation Module: It seems that the evaluation module is purely an LLM-based module, leveraging a multi-dimension scoring system like self-refine (https://arxiv.org/abs/2303.17651). I have some concerns about the module: All the feedbacks are generated by LLMs, instead of interactions with environments (e.g., runtime error, etc.). The evaluation scores generated by the module are also unreliable as well, especially when the module is following a zero-shot format. The quality of the evaluation scores will impact the framework largely.
- The framework lacks some details or implementation recommendations (How does the number of agents influence the performance? How do the tools influence the final problem-solving? How many turns on average are needed to solve the problems in the tasks displayed in the paper?)
- Some basic explanations/analyses are missing in the paper as well. (1) In Table 1, why do agents based on GPT-3.5 have better performance in a solo setting than group? Why is the situation inverse for GPT-4-based agents? How do the agents' abilities matter in the framework? (2) It seems that the Group setting performance usually has a small gain compared with CoT and Solo. I am just wondering whether Group can be easily surpassed by the other techniques, like self-refine, etc.
- The entire framework is too long and complicated. The prompts include module prompts, interaction history, and agent prompts, which are a large number of tokens to call the OpenAI API. The cost is a huge concern here to use the framework for evaluation. Furthermore, the framework is also much too inefficiency compared with the performance gain led by the multi-agent interactions.

**Questions:**

- This is such a complicated system. I am curious about the average cost of the tasks when calling GPT3.5/GPT-4 APIs as agents.
- Compared with the single agent works, what is the main motivation of using multiple agents to solve the problem? Some self-refine/self-ensemble works seem to have better performance compared with multi-agents, like (self-consistency (https://arxiv.org/abs/2203.11171), reflexion (https://arxiv.org/abs/2303.11366), self-debug (https://arxiv.org/abs/2304.05128)).

**Details Of Ethics Concerns:**

No specific ethics concerns.

---

> ### Author Response · Authors · 2023-11-15
> **Rebuttal Initial Response (1/2)**
>
> Thank you for your insightful comments regarding our methodology. We address several concerns you have raised here.
>
> ---
>
> **Q1: Concerns about the evaluation module relying solely on LLMs**
>
> A1: We agree with you that relying solely on LLMs is unreliable. We've incorporated external feedback as a key component in our evaluation module across various scenarios, enhancing reliability and effectiveness. Specifically:
>
> - **Minecraft Environment:** Evaluator receives in-game environmental factors and player states, ensuring real-time, context-specific assessment.
> - **Humaneval Benchmark:** Evaluator leverages the results from model-generated unit tests to validate outcomes, as detailed in Appendix A.
> - **Commongen Benchmark:** Evaluator uses coverage test results to gauge the effectiveness of generated responses, also elaborated in Appendix A.
> - **Tool Usage Scenarios**: Evaluator integrates outcomes from different agents to deliver accurate and comprehensive feedback or reports to users.
>
> ---
>
> **Q2: Motivation for using multiple agents**
>
> A2: As single-agent capabilities continue to evolve, a key area of research is enabling these agents to collaborate beyond solitary operation, particularly in complex real-world challenges. Our work focuses on this collaborative potential among AI agents.
>
> In single-agent research, the focus often lies on benchmark performance (similar to our Sections 3.1 and 3.2). However, the essence of multi-agent study extends beyond these metrics, delving into the autonomous behaviors and communication patterns within agent dialogues—crucial yet often overlooked aspects of agent interaction.
>
> Therefore, we do not stop at reporting higher metrics than single agent on benchmarks, we further explore the emergent behaviors. While less tangible than numerical benchmark results, **analyzing and leveraging these emergent phenomena is central to advancing multi-agent systems**. Current LLM agents are typically viewed as 'assistants' (e.g., AutoGPT, OpenAI Assistant), performing tasks as directed by humans on their own, their potential as autonomous 'agents' interacting and collaborating with one another remains largely untapped.
>
> Our research, particularly in Sections 3.3 (Tool Using) and 4 (Minecraft Game Playing), demonstrates the capabilities of LLM agents in autonomous coordination and collaborative task accomplishment. These scenarios highlight the need to consider LLM agents not just as assistants but as interactive participants in a multi-agent environment.
>
> Thus, while multi-agent systems may not always outperform single agents in certain benchmarks, their unique aspects, like emergent behaviors and communication patterns, are valuable research areas in their own right.
>
> ---
>
> **Q3: Framework complexity and cost**
>
> A3: We acknowledge the increased complexity and cost of our multi-agent system compared to single-agent systems. This complexity is inherent in our research question: "As single-agent capabilities continue to evolve, how can we enable these agents to collaborate beyond solitary operation, especially in complex real-world challenges?" Drawing parallels with human society, where collaboration leads to greater efficiency and the ability to tackle more complex tasks, our work delves into the potential and challenges of multi-agent collaborations. Indeed, we have observed inefficiencies in agent collaboration, especially agent communication inefficiency. However, as one of the pioneering studies in autonomous multi-agent systems, we explore the potential of collaborative agents, and we demonstrate it with several promising cases achieved with our framework. We acknowledge that communication challenges exist, but we argue that by highlighting these challenges, we can pave the way for future research to develop more efficient multi-agent system with advanced communication capabilities. **In light of this, we think the complexity and cost of the framework is acceptable.**
>
> If you also think this direction is worth exploring and interesting (as suggested by your strength comments), you might agree that identifying the challenging issues and providing the right direction for further research is as important as proposing incremental methods to optimize an autonomous multi-agent system. This approach allows us to highlight the great potential, as we as some limitations of agents in collaborative settings.

---

> ### Author Response · Authors · 2023-11-15
> **Rebuttal Initial Response (2/2)**
>
> **Q4: Lack of detail in experiments**
>
> A4: Thanks for your suggestion. We have carried out some additional experiments to enrich our presentation. Here we list how the number of agents affects the performance on the quantitative experiments for GPT-3.5-turbo. We run each setting on Humaneval (coding) and MGSM (math) for 3 runs, and report the averaged performance.
>
> (A brief recap, Solo means 1 role assigner agent + 1 decision-making agent + 1 evaluation agent, Group is the same except that there are multiple decision-making agent)
>
> |                        | CoT              | Solo             | Group-2          | Group-3          | Group-4          |
> | ---------------------- | ---------------- | ---------------- | ---------------- | ---------------- | ---------------- |
> | Mathematical Reasoning | $81.2_{\pm 0.6}$ | $83.2_{\pm 1.5}$ | $82.3_{\pm 0.2}$ | $82.8_{\pm 0}$   | $81.6_{\pm 1.3}$ |
> | Programming            | $72.4_{\pm 0.3}$ | $73.4_{\pm 1.9}$ | $74.8_{\pm 0.7}$ | $74.6_{\pm 1.1}$ | $74.8_{\pm 0.6}$ |
> | Average Performance    | $76.8_{\pm 0.2}$           | $78.3_{\pm 1.0}$           | $78.5_{\pm 0.5}$           | $78.7_{\pm 0.5}$           | $78.2_{\pm 0.9}$           |
>
> where `Group-x` indicates `x` decision making agent. Generally, using AgentVerse framework with one to three decision making agents give satisfying results. The averaged performance gets highest when there are 3 decision making agents. While these datasets primarily test specific agent abilities, not fully utilizing the diversity of a multi-agent setup, we still observe an upward trend in average performance with an increase in agents. The diminishing returns upon further scaling can be attributed to communication inefficiencies, as discussed in Section 3.1.
>
> At the meantime, we highlight that AgentVerse’s true potential is best observed in more complex challenges. Our case studies, tool utilization experiments, and Minecraft game-playing scenarios are prime examples where the multi-agent framework's capabilities are more pronounced and beneficial.
>
> ---
>
> Q5: Missing analyses and explanations
>
> A5: In Section 3.1, we discuss the performance variance between Group and Solo settings for GPT-3.5-turbo and GPT-4. We also examine the conflict resolution capabilities of LLMs in multi-agent systems. Further detailed explanations on these phenomena will be added to enhance understanding.

---

> > ### Comment · Reviewer_Zbb4 · 2023-11-20
> > **Official Comment by Reviewer Zbb4**
> >
> > Thank you for your efforts during the rebuttal stage. My concerns have been resolved. I will maintain my positive score and vote for acceptance.

---

> > > ### Author Response · Authors · 2023-11-23
> > >
> > > Dear reviewer Zbb4,
> > >
> > > Thank you very much for your kind response and for acknowledging our efforts during the rebuttal stage. We are heartened to hear that your concerns have been resolved, and we greatly appreciate your positive vote for acceptance.
> > >
> > > Given the additional experimental results, the clearer comparison with other frameworks, and other improvements made in our response to all the reviewers' comments, we are wondering if you might consider raising the score you have assigned.
> > >
> > > We fully respect your expertise and judgment, and understand if the current score is a reflection of your final evaluation. However, we hope that the additional efforts we have made to enhance the quality of our work across all aspects might be reflected in an updated assessment.
> > >
> > > Thank you once again for your time, consideration, and the positive impact you have had on our work. Your support and guidance throughout this process have been invaluable.

---

### Meta-Review · Area_Chair_cran · 2023-12-07

**Metareview:**

This paper proposes AGENTVERSE, a framework to coordinate a collaborative group of LLM agents. The experiments show that AGENTVERSE can efficiently organize multi-agent groups that outperform a single agent. The effectiveness of AGENTVERSE is evaluated through extensive evaluations in various tasks including text understanding, reasoning, coding, tool utilization, and embodied AI.

This is a timely work that studies a framework to organize a group of LLM agents that outperform a single LLM agent, though the technical contribution is limited. Most issues and concerns raised by the reviewers were addressed by the authors' responses. I urge the authors to incorporate necessary discussions (e.g., comparison with other frameworks) and additional experiments into the final version.

**Justification For Why Not Higher Score:**

The technical contribution is limited, as there are no new algorithms or neural architectures proposed.

**Justification For Why Not Lower Score:**

The proposed framework is extensively evaluated in various tasks. All reviewers reached a consensus on marginally above the acceptance threshold.

---

### Decision · Program_Chairs · 2024-01-16

Accept (poster)